# Bacteroid Development, Transcriptome, and Symbiotic Nitrogen-Fixing Comparison of *Bradyrhizobium arachidis* in Nodules of Peanut (*Arachis hypogaea*) and Medicinal Legume *Sophora flavescens*

Wen Feng Chen,[a,b] Xiang Fei Meng,[a,b]* Yin Shan Jiao,[a,b]§ Chang Fu Tian,[a,b] Xin Hua Sui,[a,b] Jian Jiao,[a,b] En Tao Wang,[c] Sheng Jun Ma[d]

[a]State Key Laboratory of Agrobiotechnology, Beijing, People's Republic of China
[b]College of Biological Sciences and Rhizobium Research Center, China Agricultural University, Beijing, People's Republic of China
[c]Departamento de Microbiología, Escuela Nacional de Ciencias Biológicas, Instituto Politécnico Nacional, México City, México
[d]College of Food Science and Pharmacy, Xinjiang Agricultural University, Urumqi, Xinjiang Uygur Autonomous Region, People's Republic of China

**ABSTRACT** *Bradyrhizobium arachidis* strain CCBAU 051107 could differentiate into swollen and nonswollen bacteroids in determinate root nodules of peanut (*Arachis hypogaea*) and indeterminate nodules of *Sophora flavescens*, respectively, with different $N_2$ fixation efficiencies. To reveal the mechanism of bacteroid differentiation and symbiosis efficiency in association with different hosts, morphologies, transcriptomes, and nitrogen fixation efficiencies of the root nodules induced by strain CCBAU 051107 on these two plants were compared. Our results indicated that the nitrogenase activity of peanut nodules was 3 times higher than that of *S. flavescens* nodules, demonstrating the effects of rhizobium-host interaction on symbiotic effectiveness. With transcriptome comparisons, genes involved in biological nitrogen fixation (BNF) and energy metabolism were upregulated, while those involved in DNA replication, bacterial chemotaxis, and flagellar assembly were significantly downregulated in both types of bacteroids compared with those in free-living cells. However, expression levels of genes involved in BNF, the tricarboxylic acid (TCA) cycle, the pentose phosphate pathway, hydrogenase synthesis, poly-$\beta$-hydroxybutyrate (PHB) degradation, and peptidoglycan biosynthesis were significantly greater in the swollen bacteroids of peanut than those in the nonswollen bacteroids of *S. flavescens*, while contrasting situations were found in expression of genes involved in urea degradation, PHB synthesis, and nitrogen assimilation. Especially higher expression of *ureABEF* and *aspB* genes in bacteroids of *S. flavescens* might imply that the BNF product and nitrogen transport pathway were different from those in peanut. Our study revealed the first differences in bacteroid differentiation and metabolism of these two hosts and will be helpful for us to explore higher-efficiency symbiosis between rhizobia and legumes.

**IMPORTANCE** Rhizobial differentiation into bacteroids in leguminous nodules attracts scientists to investigate its different aspects. The development of bacteroids in the nodule of the important oil crop peanut was first investigated and compared to the status in the nodule of the extremely promiscuous medicinal legume *Sophora flavescens* by using just a single rhizobial strain of *Bradyrhizobium arachidis*, CCBAU 051107. This strain differentiates into swollen bacteroids in peanut nodules and nonswollen bacteroids in *S. flavescens* nodules. The $N_2$-fixing efficiency of the peanut nodules is three times higher than that of *S. flavescens*. By comparing the transcriptomes of their bacteroids, we found that they have similar gene expression spectra, such as nitrogen fixation and motivity, but different spectra in terms of urease activity and peptidoglycan biosynthesis. Those altered levels of gene expression might be related to their functions and differentiation in respective nodules. Our

Address correspondence to Wen Feng Chen, chenwf@cau.edu.cn.

*Present address: Xiang Fei Menga, CSPC Ouyi Pharmaceutical Co., Ltd., Shijiazhuang, Hebei Province, People's Republic of China.

§Present address: Yin Shan Jiao, Dazu Yunfeng, Longgang District, Shenzhen City, Guangdong Province, People's Republic of China.

The authors declare no conflict of interest.

studies provided novel insight into the rhizobial differentiation and metabolic alteration in different hosts.

**KEYWORDS** *Bradyrhizobium arachidis*, peanut, swollen bacteroid, nonswollen bacteroid, *Sophora flavescens*, bacteroid differentiation, root nodule, legume, rhizobia, *Arachis hypogaea*, RNA-Seq, differentiation, symbiotic nitrogen fixation, transcriptome

Peanut (*Arachis hypogaea*) is an important oil crop widely grown globally, and it establishes determinate root nodules for biological nitrogen fixation (BNF) with the genus *Bradyrhizobium*, including *Bradyrhizobium arachidis* (1) and several other bradyrhizobial species (2, 3). Inside its root nodules, the bradyrhizobia differentiate into bacteroids with spherical and swollen morphotypes (4, 5). The medicinal legume *Sophora flavescens* (6–8) is an extremely promiscuous host, establishing symbioses with more than 35 rhizobial species in both of the classes *Alphaproteobacteria* and *Betaproteobacteria* (9–11), including the currently studied strain CCBAU 051107 of the peanut microsymbiont *B. arachidis* (9, 12). Because of its distinct promiscuity, *S. flavescens* might be used as an excellent plant to study the development and physiology of various rhizobia and bacteroids inside its indeterminate root nodules (11). We previously found that the Rlv3841 of *R. leguminosarum* bv. *viciae* underwent characteristically nonterminal and terminal differentiations in root nodules of *S. flavescens* and pea (*Pisum sativum*), respectively (11). Furthermore, we observed sphere-like bacteroids inside root nodules of *S. flavescens* infected by strain NZP 2213 of *Mesorhizobium loti* (9). These studies mentioned above demonstrated that the differentiation of rhizobia to bacteroids in root nodules was regulated by both of the symbiotic partners. However, the regulatory mechanism and effects of different shapes of bacteroids on symbiosis compatibility and nitrogen fixation are still not clear.

Based on our knowledge, few studies have ever compared the transcriptomes of bacteroids inside the determinate and indeterminate root nodules induced by the same rhizobial strain. Considering a comparative study of the development, physiology, and metabolism of bacteroids generated by the same rhizobial strain to be essential for understanding the symbiotic interactions between rhizobia and legumes, we performed the present study. The aim of this study was to reveal the mechanism of bacteroids' differentiation and their efficiency at symbiosis in association with different hosts. The results offered us interesting information in the field of symbiotic interaction between rhizobia and legumes.

## RESULTS

**Symbiotic characterization of root nodules.** Strain CCBAU 051107 of *B. arachidis* induced determinate nodules on peanut and indeterminate nodules on *S. flavescens*, with a dark red color and light red color, respectively, in their sections (Fig. 1A1, B1, A2, B2). The total numbers of root nodules per plant of peanut, 34.0 ± 2.3 (mean ± standard deviation [SD]), were significantly higher than those of *S. flavescens* (17.3 ± 2.0) ($P < 0.05$) (Fig. 1A1, B1). The diameter of mature peanut root nodules was about 1 mm, while the size of root nodules of *S. flavescens* was 2 mm in length by 1 mm in width.

The root nodules of peanut had a spherical meristem zone (SMZ), while those of *S. flavescens* had an apical meristem zone (AMZ) (Fig. 1B3). The nitrogen fixation (NF) zone of peanut nodule (Fig. 1A3) was smaller than that of the *S. flavescens* nodule (Fig. 1B3), corresponding to the red color ranges in their responding nodule sections (Fig. 1A2 and Fig. 1B2), while the former had a higher and homogeneous density of bacteroids (Fig. 1A3, a few white plaques) than the latter (Fig. 1B3, many white plaques). Bacteroids inside the peanut root nodule contained polyphosphate (PP) inclusions but no poly-$\beta$-hydroxybutyrate (PHB) particles (Fig. 1A4), while bacteroids inside nodules of *S. flavescens* had obvious PHB particles but no PP (Fig. 1B4). Only one bacteroid was enclosed in each symbiosome in the peanut nodule cell with no peribacteroid space (PBS), while multiple bacteroids (up to 10 or more) were housed in the symbiosome of *S. flavescens* nodule cell with a distinct PBS (Fig. 1B4). The diameter (Φ) of the spherical peanut bacteroids was 1.338 ± 0.083 $\mu$m,

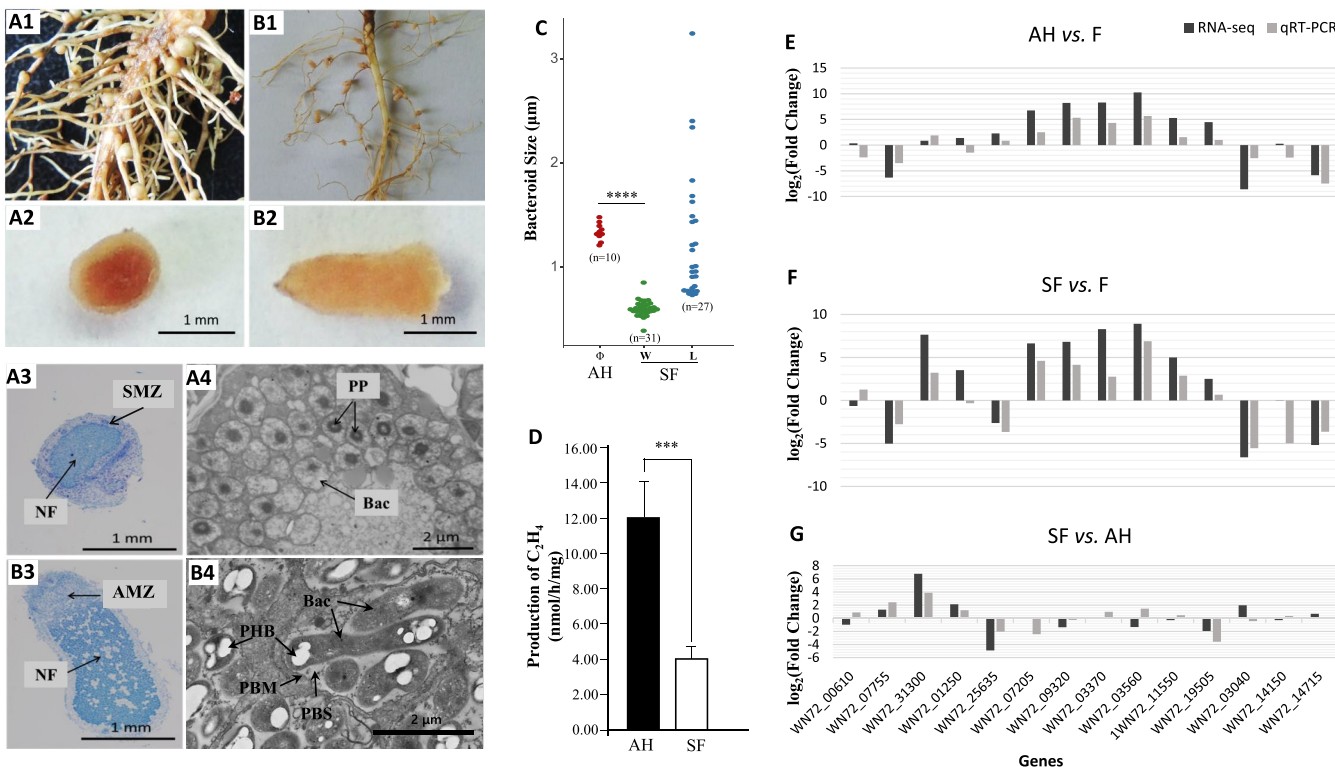

**FIG 1** Symbiotic characterization and validation of RNA-Seq versus qRT-PCR. (A and B) Root system, root nodule, nodule section, and transmission electron microscopy (TEM) images of peanut (*A. hypogaea*) (A1 to A4) and *S. flavescens* (B1 to B4). The sizes of bacteroids (C) were determined using the software ImageJ, and panel C was drawn by using the *ggbeeswarm* package in the R language. The acetylene reduction assay is represented by the level of production of the $C_2H_4$ of root nodules of peanut (*A. hypogaea*) and *S. flavescens* (D). (E to G) Validation of RNA-Seq results by qRT-PCR using a total of 14 genes. NF, nitrogen fixation zone; AMZ, apical meristem zone; SMZ, spherical meristem zone; PP, electron-opaque polyphosphate inclusions; Bac, bacteroid; PBM, peribacteroid membrane; PBS, peribacteroid space; PHB, poly-$\beta$-hydroxybutyrate. Asterisks (*** and ****) above the dots or bars indicate the significant difference between the two samples of bacteroids of peanut (*A. hypogaea* [AH]) and *S. flavescens* (SF) using the *t* test. F, free-living rhizobia.

ranging from 1.209 to 1.478 $\mu$m, and they were significantly wider than width (W) of the rod bacteroids in *S. flavescens* nodules, which presented an average of 0.603 $\pm$ 0.076 $\mu$m in width (W) (Fig. 1C) ($P < 0.0001$). Comparatively, the length (L) of rod bacteroids in *S. flavescens* nodules changed a lot, from 0.730 to 3.243 $\mu$m in length (L) (Fig. 1C). In addition, the average area of each peanut bacteroid was 1.492 $\pm$ 0.202 $\mu$m², and the area was only 0.625 $\pm$ 0.307 $\mu$m² in *S. flavescens* bacteroids. The nitrogenase activity of peanut nodules was 12.06 $\pm$ 2.04 nmol/h/mg, which was about 3 times higher than that nodules of *S. flavescens* (3.81 $\pm$ 0.58 nmol/h/mg) (Fig. 1D).

**RIN, sequencing data, and mapping to known genome sequence.** The RNA integrity number (RIN) values of RNA samples were above 7.5, meeting the high-quality requirements for transcriptome sequencing (RNA-Seq) library construction. RNA samples were sent to the Beijing Huada Gene Company to construct the library and for subsequent sequencing. The sequence data were mapped to the genome of strain CCBAU 051107. Total clean reads of each sample were more than 10 Mb, and the read length of double-terminal sequencing was 100 bp, so the total base number of each sample was more than 1 Gb. The mapping ratios of total clean reads on the genome sequence of the strain CCBAU 051107 were more than 90% for free-living cells, 28% for peanut bacteroids, and 24% for *S. flavescens* bacteroids. The sequencing depth of bacteroids (200 Mb) reached more than 30 times the genome size (7 Mb), which fully met the requirements.

**Verification of gene expression by qRT-PCR and general comparison.** A total of 14 genes (Fig. 1E to G) were selected for real-time quantitative PCR (qRT-PCR) verification of the RNA-Seq results, and good consistency was observed between them, with a Pearson correlation coefficient of 0.853 ($P < 0.0001$) and linear $R^2$ value of 0.727, which proved the accuracy of the RNA-Seq results.

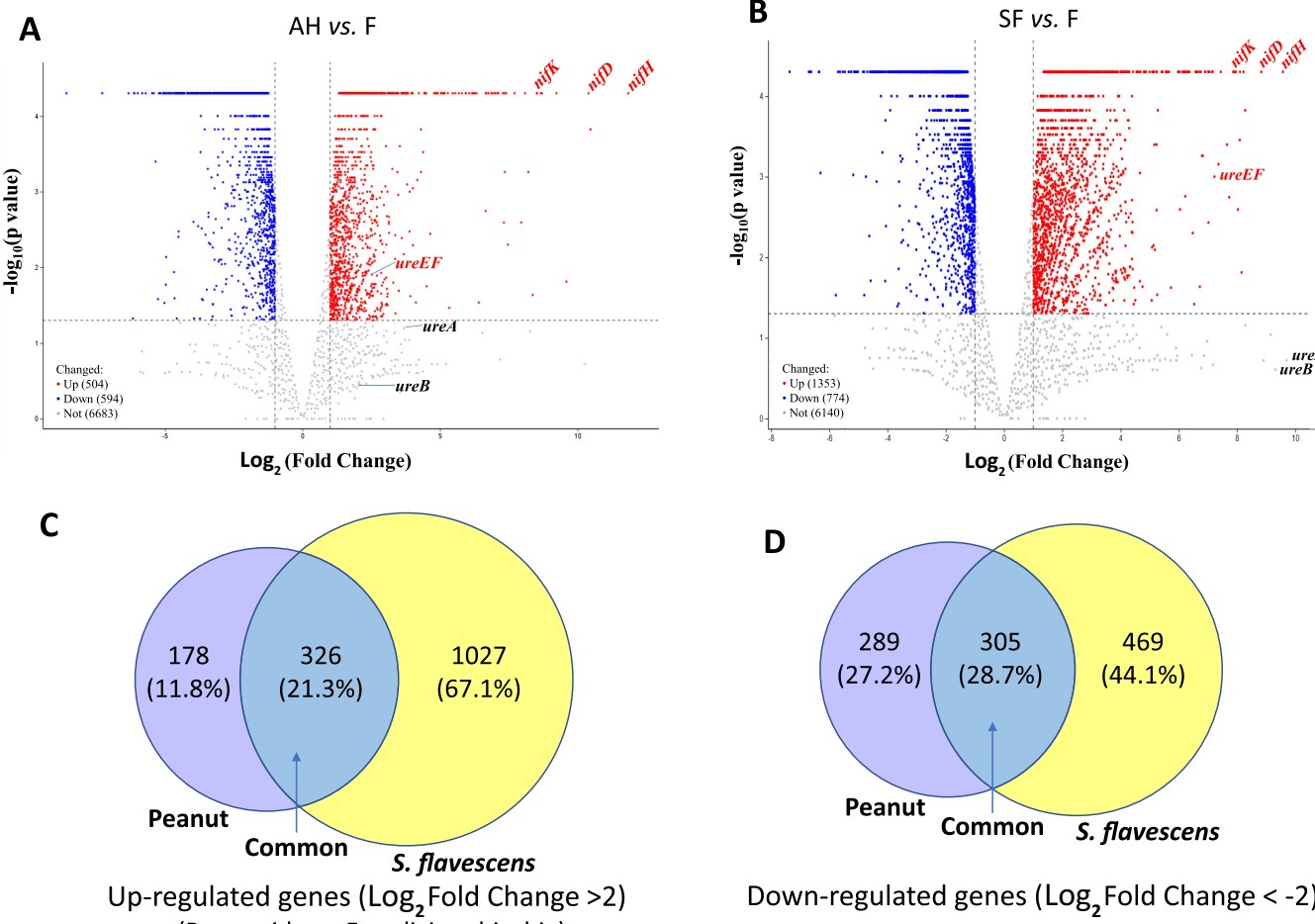

**FIG 2** Volcano plots (A and B) and Venn diagrams (C and D) showing the upregulated (C) and downregulated (B) expression of genes in bacteroids of peanut (*A. hypogaea* [AH]) and *S. flavescens* (SF) versus free-living rhizobia (F). Selected differentially upregulated expressed genes (*nifHDK* and *ureABEF*) are labeled beside their dots in panels A and B. The Venn diagrams were constructed using the online tool of Venny v.2.1.0 (69). The volcano plots were constructed by using function *ggscatter* of the *ggpubr* package in the R language.

In comparison with free-living cells, 1,046 and 1,590 differentially expressed genes (DEGs) were found in bacteroids of peanut nodules (including 504 upregulated genes and 594 downregulated genes) (Fig. 2A) and in bacteroids of *S. flavescens* nodules (including 1,353 upregulated genes and 774 downregulated genes) (Fig. 2B), respectively. Of these upregulated genes, those coding for components of nitrogenase (*nifHDK*) ranked first among the top-ranked genes in both kinds of bacteroids (Fig. 2A and B). As DEGs, the ure-ase coding genes *ureABEF* were highlighted in bacteroids of *S. flavescens*, with log$_2$ fold change (FC) values of 9.69, 9.31, 7.22, and 7.22, respectively (Fig. 2B), while the expression levels of *ureABE* genes in peanut bacteroids were not as high, with log$_2$ FC values of 3.81, 2.82, and 2.52, respectively, compared with those of free-living rhizobial cells (Fig. 2A).

As shown in the Venn diagram (Fig. 2C and D), the numbers of upregulated genes specifically for bacteroids in *S. flavescens* (1,027 [67.1%]) were higher than those in pea-nut (178 [11.8%]), and their common gene numbers were 326 (21.3%). Comparatively, the numbers of common and specific downregulated genes were almost similar, with 305 (28.7%) commonly downregulated genes and 289 (27.2%) and 469 (44.1%) specifi-cally downregulated genes in bacteroids of peanut and *S. flavescens*, respectively (Fig. 2D). Overall, the total numbers of DEGs in bacteroids of *S. flavescens* were higher than those in bacteroids of peanut (Fig. 2C and D).

**COG analysis of DEGs.** The results of COG (Clusters of Orthologous Groups) analysis shown in Fig. 3 demonstrated that the significantly changed DEGs, either upregulated

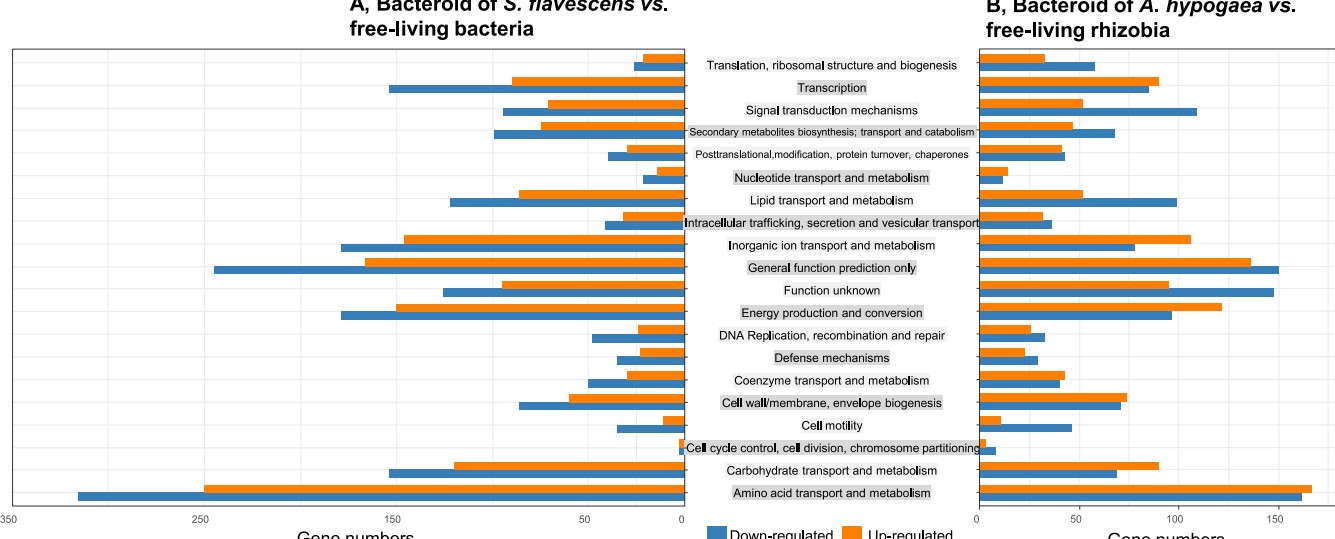

**FIG 3** Distributions and numbers of differentially expressed genes (DEGs) in COG functional categories. (A) Bacteroids of *S. flavescens* versus free-living bacteria; (B) Bacteroids of *A. hypogaea* versus free-living rhizobia. Gene expression levels were downregulated or upregulated, respectively, in the two kinds of bacteroids and were classified into different COG categories compared with the free-living bacteria. Numbers of DEGs were counted. The plot was constructed using the functions *ggplot* and *ggarrange* of the *ggplot2* and *ggpubr* packages in the R language.

or downregulated, were mainly involved in amino acid transport and metabolism, energy production and conversion, carbohydrate transport and metabolism, transcription, and inorganic transport and metabolism, besides the unknown function and predicted general function genes. The downregulated genes mainly participated in cellular processes and signaling, cell motility, DNA replication, cell division and the chromosome, as well as genes related to translation, ribosomal structure, and biogenesis. The trends of these DEGs in two kinds of bacteroids were generally consistent. However, the total numbers of DEGs in the COG category of amino acid transport and metabolism in *S. flavescens* bacteroids (Fig. 3A) were twice as high as those in peanut bacteroids (Fig. 3B).

**DEGs matched in the KEGG pathway database.** The metabolic pathways involving the upregulated and downregulated DEGs, estimated by using the KEGG (Kyoto Encyclopedia of Genes and Genomes) database, are shown in Fig. 4 and Fig. 5. The first 30 genes with upregulated and downregulated expression in bacteroids of the peanut and *S. flavescens* were compared, respectively, with those in the free-living bacterial cells (Fig. 4A and B). Clearly, most of the upregulated genes (*fix* and *nif*) were involved in BNF, and of the 30 upregulated genes, 12 were common in expression profiles of the bacteroids from both plants. Of the 30 genes presenting downregulated expression, 10 were common in bacteroids of both plants and 7 were unknown hypothetical protein (HP) genes (Fig. 4A and B).

The expression levels of genes involved in chemotaxis, motility, and flagellin were downregulated in both kinds of bacteroids (Fig. 5A), such as the genes *mcp*, *cheA*, *cheR*, *cheZ*, and *motA*. In addition, the expression of these genes in peanut bacteroids was much more downregulated than the expression of those in bacteroids of *S. flavescens* (Fig. 5A).

Genes in the *exo* cluster are responsible for exopolysaccharide (EPS) synthesis. In comparison with the free-living rhizobial cells, only the *exoZ* gene presented upregulated expression, and the *exoT* and *exoU* genes showed downregulated expression, while the expression of other *exo* genes was not significantly changed in peanut bacteroids (Fig. 5B). Differently, the two genes *exoM* and *exoQ* were upregulated significantly, but the *exoZ* gene was downregulated significantly, and the expression levels of other *exo* genes were not significantly changed in bacteroids of *S. flavescens*.

Different expression levels of genes involved in lipopolysaccharide (LPS) synthesis (*lpx*) are shown in Fig. 5B. Comparatively, the expression of *lpxK* was upregulated in peanut bacteroids and that of *lpxB* was upregulated in bacteroids of *S. flavescens*, while

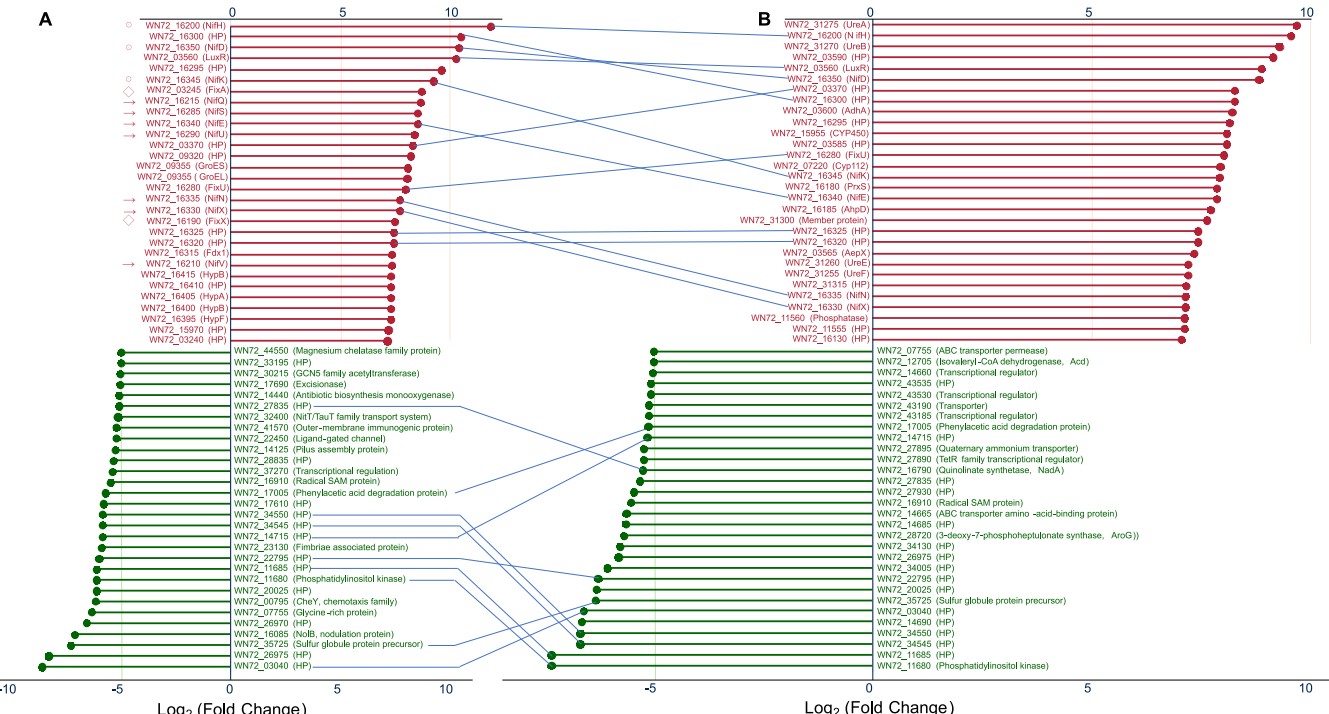

**FIG 4** Top 30 upregulated and downregulated expression genes in bacteroids of *A. hypogaea* (A) and *S. flavescens* (B) versus free-living bacteria. The genes with upregulated expression are arranged in the upper half of the figure. The downregulated genes are shown in the bottom half of the figure. Circles with a center dot represent components of nitrogenase, diamonds represent the regulators of nitrogenase activity, and arrows represents cofactor. Common genes found in two kinds of bacteroids are linked using lines. HP, hypothetical protein. The plot was constructed by using the function *diverging_lollipop_chart* of the *ggcharts* package in the R language.

the expression patterns of other *lpx* genes in both bacteroids were similar to those of the free-living rhizobial cells (Fig. 5B).

The expression levels of *nif* and *fix* genes responsible for BNF were significantly increased in bacteroids of both peanut and *S. flavescens* (Fig. 5C). The expression of structure genes for nitrogenase (*nifHDK*) ranked in the first three top DEGs among the BNF-related genes, while the expression level of the regulatory gene *nifA* was relative lower than those of the other BNF genes (log$_2$ FCs of 0.65 and 0.51, respectively, for bacteroids of peanut and *S. flavescens*). However, the absolute expression counts of *nifA* were in the following order: peanut bacteroids (1,039) > *S. flavescens* bacteroids (940) > free-living rhizobial cells (662). The expression level of another regulatory gene, *rpoN* (encoding RNA polymerase sigma factor 54 and controlled by the master regulator NifA), was highly upregulated in the peanut bacteroids but downregulated in bacteroids of *S. flavescens* (Fig. 5C). In addition, the expression levels of most *nif* and *fix* genes in bacteroids were higher in peanut nodules than those in *S. flavescens* nodules (Fig. 5C).

In nodules, the fixed ammonia was assimilated into amino acids by glutamine synthetase (GS) (encoded by *glnA* or *glnII*) or asparagine synthetase (AS) B (encoded by *asnB*) (13–15). With the exception of the *glnA* gene (WN72_09535), whose expression was downregulated in bacteroids of both host plants, the expression of the *glnII* gene (WN72_10230) in peanut bacteroids and the *asnB* gene (WN72_21080) in bacteroids of *S. flavescens* was upregulated (Fig. 5D) (both log$_2$ FCs of ≥2 and corrected *P* value [*q*] of <0.05), respectively.

Nodulation genes mainly included gene clusters of *nod*, *nol*, and *noe*. The expression levels of *nod* genes (except *nodM* and *nodJ*) in both bacteroids were not significantly different from those in the free-living rhizobia, while the expression levels of *nolABOUY* genes in both bacteroids were significantly downregulated (Fig. 5E). The expression of the *noeI* gene was slightly upregulated in peanut bacteroids. There was no significant difference in expression of structure genes for nodulation factor

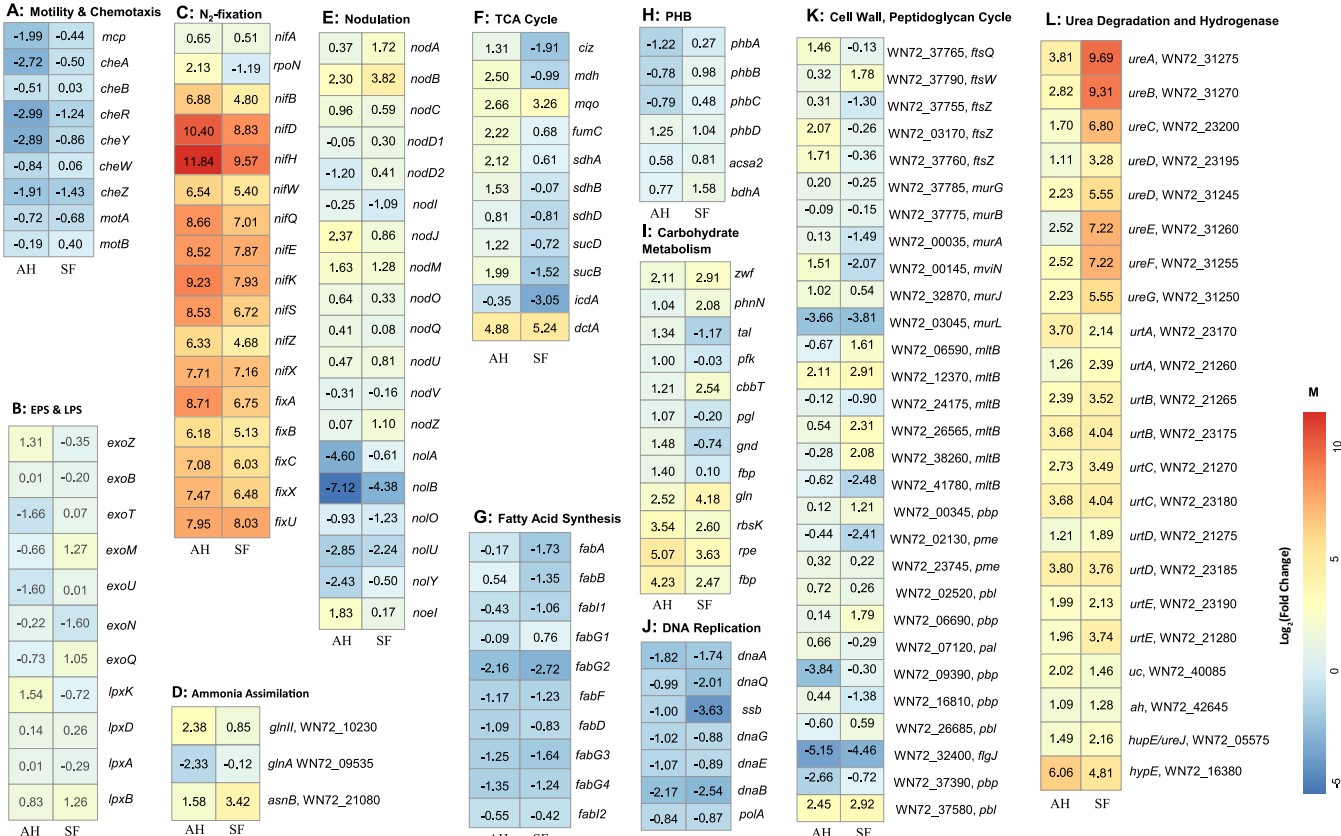

**FIG 5** Differentially expressed genes (DEGs) matched in the KEGG database in different pathways displayed as a heat map. The data were from the log₂ FCs of DEGs in bacteroids of *A. hypogaea* (AH) versus free-living bacteria and bacteroids of *S. flavescens* (SF) versus free-living bacteria. The color key is shown in panel M. The plot was constructed by using the function *pheatmap* of the *pheatmap* package in the R language. The plot was transferred into PowerPoint by using the function *topptx* of the *eoffice* package in the R language to revise it manually.

synthesis (*nodABC*) between the two kinds of bacteroids, while the expression of regulation genes (*nodD1* and *nodD2*) changed a lot, especially for the *nodD2* gene, which was downregulated significantly in bacteroids of peanut.

The expression of the *dctA* gene, responsible for the transport of tetracarboxylic acid to bacteroids as a carbon source, was highly upregulated in both bacteroids (Fig. 5F), with FPKM (fragments per kilobase per million) values ranging from 17 to 513 and 657, respectively. Most genes involved in tricarboxylic acid (TCA) cycle (Fig. 5F) were significantly upregulated in the peanut bacteroids, including *ciz* (citrate synthase), *mdh* (malate dehydrogenase), *mqo* (malate:quinone oxidoreductase), *fumC* (fumarase), *sdhABD* (succinate dehydrogenase), and *sucDB* (succinyl coenzyme), while the expression levels of these genes and *icdA* (isocitrate dehydrogenase) were significantly lower in bacteroids of *S. flavescens* (Fig. 5F) than those in peanut bacteroids.

The genes participating in fatty acid metabolism in both bacteroids were not active, or their expression was downregulated (Fig. 5G). The expression of structural genes (*phbABC*) involved in PHB synthesis in bacteroids of peanut was significantly downregulated, while the expression of the *phbD* gene (PHB depolymerase) increased compared to that in the free-living rhizobia, and the expression level of the *phbD* gene in peanut bacteroids was a little bit higher than that in *S. flavescens* (Fig. 5H). No significant difference was found in gene expression of *ascA2* (acetyl coenzyme A [acetyl-CoA] synthetase) and *bdhA* (3-hydroxybutyrate dehydrogenase), central enzymes to PHB degradation, in both bacteroids (Fig. 5H).

The genes involved in the pentose phosphate pathway were active and upregulated significantly in peanut bacteroids (Fig. 5I). Comparably, the expression of these corresponding genes in bacteroids of *S. flavescens* was not so much higher, and it was

even downregulated for the genes *tal* (glucose-6-phosphate isomerase), *pfk* (6-phosphofructokinase), *pgl* (6-phosphogluconolactonase), and *gnd* (6-phosphogluconate dehydrogenase) (Fig. 5I).

The expression of genes coding for DNA replication-related proteins, replication initiation enzyme (*dnaA*), single-strand binding protein (*ssb*), DNA helicase (*dnaB*), and DNA polymerase subunits (*dnaE* and *dnaQ*) was downregulated in both bacteroids (Fig. 5J). The expression of genes of large (*rpl*) and small (*rps*) RNA subunits involved in ribosome syntheses was also downregulated in both bacteroids.

Expression levels of genes involved in cell division or peptidoglycan or murein cycles in free-living rhizobial cells and in bacteroids were significantly different (Fig. 5K). In peanut bacteroids, genes with significantly upregulated expression ($q < 0.05$) include those encoding FtsQ, FtsZ, MviN, MurJ, MltB (lytic murein transglycosylase [WN72_12370]), and Pbl (LysM peptidoglycan-binding domain-containing protein [WN72_37580]) (Fig. 5K), while genes with significantly downregulated expression ($q < 0.05$) include those encoding MurL (murein L,D-transpeptidase [WN72_03045]), Pbp (peptidoglycan-binding protein [WN72_09390 and WN72_37390]), and FlgJ (WN72_32400).

Comparatively, expression of many genes involved in cell division or peptidoglycan or murein cycles (Fig. 5K) was decreased or not significantly changed in bacteroids of *S. flavescens*, except for the genes encoding FtsW (WN72_33790), MltB (lytic murein transglycosylase [WN72_06590, WN72_26565, WN72_12370, and WN72_38260]), Pbp (peptidoglycan-binding protein [WN72_00345 and WN72_06690]), and Pbl (LysM peptidoglycan-binding domain-containing protein [WN72_37580]). Extremely significant differences ($q < 0.001$) between the bacteroids of the two hosts were observed in genes encoding MviN (WN72_00145) and FtsZ (WN72_37760 and WN72_03170) (significantly upregulated in peanut nodules) and in two Pbp (peptidoglycan-binding proteins [WN72_09390 and WN72_37390]) (significantly downregulated in peanut nodules).

In strain CCBAU 051107, complete gene clusters responsible for urease activity (*ureABC*), assembly (*ureDEFG*), and urea transport (*urtABCDE*) could be detected in the transcriptome, and most of them had two copies (Fig. 5L). Significantly, the expression level of urea degradation-related genes was higher in bacteroids of *S. flavescens* than that in peanut bacteroids, especially for the genes *ureABCDFG* and *urtBCE*. In addition, the expression levels of the genes coding for urea carboxylase (*uc*) and allophanate hydrolase (*ah*) were also detected in both kinds of bacteroids (Fig. 5L).

The expression level of the *hypE* gene, encoding HypE (hydrogenase expression/formation protein), involved in Ni-Fe hydrogenase biosynthesis in the peanut bacteroids, was extremely higher than that in bacteroids of *S. flavescens* (Fig. 5L). Expression of genes for the accessory proteins of hydrogenase/urease (*hupE*/*ureJ*) was upregulated in both kinds of bacteroids (Fig. 5L).

## DISCUSSION

Previously, a comparative study of symbiosis of dual-host *Bradyrhizobium* sp. strain 32H1 with the peanut and the cowpea, both with formed determinate nodules, revealed that the swollen bacteroids in peanut nodules conferred more net host benefit in terms of nodule construction cost (plant growth per gram of nodule growth) and BNF efficiency (5). The results in our present study support this previous observation, since the nitrogenase activity of peanut nodules was about 3 times greater than that of *S. flavescens*, and it was consistent with the transcriptomic data since the expression levels of N₂-fixing genes were 2 to 5 times greater in the bacteroids of peanut nodules than those in bacteroids of *S. flavescens*. Differing from the previous study (5), the root nodules of the peanut and *S. flavescens* were determinate and indeterminate, respectively, in our study. The differences in nitrogen fixation between the swollen and nonswollen bacteroids found in our present study and in the previous study (5) might be related to the symbiotic compatibility between rhizobial strains and the legume species (16, 17), while the differentiation of swollen/nonswollen bacteroids might be regulated by the host species. Previously, it was described that transcriptomes of *Sinorhizobium* sp. strain NGR234 could form determinate nodules on *Vigna unguiculata* and

indeterminate nodules on *Leucaena leucocephala*, which differentiated into only the non-swollen type of bacteroids, and only one bacteroid was encapsulated for each symbiosome in the nodules of both of the host plants (18). In our present study, the distinct characteristic of strain CCBAU 051107 was that it differentiated into swollen bacteroids and only one bacteroid was encapsulated by a peribacteroid membrane in the peanut nodule cells, while it differentiated into nonswollen bacteroids and multiple bacteroids were encapsulated in peribacteroid membrane in *S. flavescens* nodule cells.

Through the comparison of COG functional categories, the DEGs in bacteroids were predominantly involved in amino acid transport and metabolism, clearly consistent with the primary N$_2$ fixation role of bacteroids in the nodules (19, 20). Our present results indicated that the functional classifications of DEGs (either upregulated or downregulated) were very similar in both swollen and nonswollen bacteroids. However, comparisons of various metabolic pathways by using the KEGG database confirmed the metabolic differences between the two kinds of bacteroids. The expression levels of genes related to functions of chemotaxis, motility, and flagellin were downregulated in both kinds of bacteroids, which was consistent with the previous report that the expression of many genes became undetectable in the differentiated bacteroids inside mature root nodules (21). The expression of gene WN72_32400, encoding flagellar assembly peptidoglycan hydrolase (FlgJ), declined dramatically in both kinds of bacteroids, further confirming the motionless status specializing in BNF inside the root nodules. Since the protein FlgJ was essential for flagellar rod formation in the Gram-negative bacteria *Salmonella enterica* serovar Typhimurium and *Brucella abortus* (22–24), the decline in expression of the *flgJ* gene would cause the loss of flagellar and moving function in the N$_2$-fixing bacteroids inside the root nodules.

Besides the differences in BNF efficiency, expression levels of genes involved in the nitrogen assimilation (glutamine synthetases [GSs] and asparagine synthetases [ASs]), urease activity, nitrogen-fixing regulator (NifA and RpoN), and hydrogen uptake were also different, confirming the distinction between these two kinds of bacteroids and their distinctive nitrogen metabolism pathways. GSs are key enzymes of nitrogen assimilation, and four forms of GSs have been described in the three life domains: *Bacteria*, *Archaea*, and *Eukaryota* (25, 26). Two kinds of GSs (GSI encoded by *glnA*, and GlnII encoded by *glnII*) were found in strain CCBAU 051107, the same as another reported rhizobium (27). In present study, upregulated expression of the *glnII* gene and downregulated expression of *glnA* were detected in both kinds of bacteroids, implying GlnII might be functioning in symbiosis for transport of the fixed nitrogen (NH$_3$) from bacteroids to the host plants, while GlnA might be the ammonium transporter responsible for amino acid synthesis in free-living cells of strain CCBAU 051107. The detection of a greater expression level of *glnII* in peanut bacteroids than in bacteroids of *S. flavescens* might be a characteristic related to the N$_2$-fixing efficiency or compatibility of the two symbiosis combinations.

Another important enzyme in nitrogen assimilation is the asparagine synthetase (AS) (14) encoded by *asnB* (glutamine-dependent) or *asnA* (NH$_4^+$-dependent). In the present study, *asnB* was detected as one of the DEGs in both kinds of bacteroids, while its expression in bacteroids of *S. flavescens* was significantly greater than that in the peanut bacteroids. The differences in nitrogen assimilation in these two kinds of bacteroids meant that the primary assimilation product from N$_2$ fixation was glutamine in peanut nodules, while it perhaps was asparagine in *S. flavescens* nodules. Previously, it was reported that asparagine was the predominant nitrogen transport product in alfalfa (14, 28), while in soybean nodules, the GS/GOGAT (glutamine synthetase/glutamine oxoglutarate aminotransferase) system was the main ammonia assimilation pathway (29). Our results provided further evidence that the transport pathway of the fixed nitrogen was dependent to the host plants.

Surprisingly, the *ureA* gene, encoding urease, ranks no. 1 in the top 30 upregulated genes in bacteroids of *S. flavescens*, even higher than that *nifH* (which ranks no. 2), one of the most important BNF genes. The expression levels of other urease-related genes (*ureBC* and *ureF* [WN72_31255]) were also upregulated in bacteroids of *S. flavescens*, while these *ure* genes were not listed in the top 30 upregulated DEGs in peanut

bacteroids, meaning that their expression levels were relatively lower in peanut nodules. Urease is one of the important enzymes in the nitrogen cycle (30, 31), catalyzing the hydrolysis of urea to yield ammonia and carbamate (32, 33). Urea can be utilized as a nitrogen source by some rhizobia (34), the marine sponge *Xestospongia testudinaria* (35), and cyanobacteria (31). In the genome of strain CCBAU 051107, all of the urea-related genes (*ureABCDEFG*, *urtABCDE*, *uc*, and *ah*) could be detected, similarly to the composition of some other nitrogen-fixing cyanobacteria (31). Based on the greatly upregulated expression of urea degradation genes in the bacteroids of *S. flavescens* in the present study, we estimated that urea might be the intermediate metabolite of nodule cells and could be degraded by symbiotic bacteroids to ammonia and carbon dioxide, another distinct nitrogen cycle. High expression of the *ureA* gene in bacteroids of *Phaseolus vulgaris* infected by *Paraburkholderia phymatum* was also reported, and it ranked in the top 200 DEGs (36). In addition, this phenomenon might be related to the transfer of fixed N from bacteroids to the plant root to synthesize (oxy)matrine, the active medicinal ingredient in *S. flavescens*, since rhizobial inoculation could increase significantly the synthesis of (oxy)matrine in this plant root (12). Both of these hypotheses remain to be further checked in the future.

Nickel is an important element for the enzymatic activity of urease and hydrogenase (34, 37). The upregulated expression levels of WN72_31280 (nickel transporter) were consistent with the upregulated expression of *ureF* (urease accessory protein [WN72_31255]) and WN72_16405 (hydrogenase nickel incorporation protein [HypA]) in bacteroids of *S. flavescens*. However, the expression levels of genes WN72_16405 (HypA), WN72_16400 (HypB), WN72_44375 (nickel responsive regulator), and WN72_16380 (hydrogenase [HypE]) were much higher in peanut bacteroids than those in *S. flavescens* bacteroids, meaning that the nickel uptake in peanut nodules was more important and might be consistent with the higher accumulation of nickel in peanut seeds (38).

The significant upregulation of gene expression related to the TCA cycle and pentose phosphate metabolism in peanut bacteroids compared to that of *S. flavescens* bacteroids displayed the difference in carbon and energy-producing metabolisms between them. The results were consistent with the activity of genes involved in the TCA cycle and pentose phosphate metabolism, further confirming the higher requirement for energy for nitrogen fixation of swollen bacteroids (39). The genes involved in fatty acid metabolism in both kinds of bacteroids were not active or downregulated. It could be speculated that the synthesis of fatty acids is a process of energy consumption, which may compete with the requirement for the nitrogen fixation in bacteroids. Therefore, fatty acid synthesis genes were not active in bacteroids. Previous studies have shown that the synthesis of fatty acids in bacteroids of *Bradyrhizobium diazoefficiens* USDA110 drastically declined (39). The bacteroid requirement for lipids might be lower because of its nondividing nature or because the bacteroids were supplied by the host plant (39).

Apparent PHB accumulation was found in bacteroids of *S. flavescens* nodules, but few were observed in bacteroids of peanut nodules, which was consistent with the up- or downregulated gene expression of PHB syntheses and degradation in these two kinds of bacteroids, respectively. Previous results have demonstrated that PHB accumulation in bacteroids was variable between rhizobial strains associated with different host plants (40, 41) or even with same host (42). Our results also implied the possibility that the PHB accumulation might decrease the efficiency of nitrogen fixation, as analyzed by Wang et al. (43) in nodules of *Medicago truncatula*. Therefore, both the nitrogen metabolism and PHB accumulation in bacteroids are regulated by the rhizobial strain and the host legume.

In a previous study, Li et al. (18) reported that the expression of many genes involved in cell division was downregulated in bacteroids of *V. unguiculata* and *L. leucocephala*, consistent with the gene expression pattern in bacteroids of *S. flavescens*, and all of these three plant species contained nonswollen bacteroids. However, some of these genes were upregulated significantly in the swollen bacteroids of peanut, especially those encoding MurJ

(MviN) and FtsZ (WN72_33760 and WN72_03170). MurJ was an inner membrane protein in Gram-negative cells responsible for murein/peptidoglycan biosynthesis by transferring lipid II into the periplasm from the cytoplasm (44, 45). MviN was required for maintenance of cellular shape and integrity (46). The role of MurJ (MviN) in development and differentiation of rhizobia into swollen bacteroids in peanut nodules remains to be elucidated in the future. However, these differences implied that the cell division genes in rhizobia might be also related to the bacteroid differentiation, and the host plant might affect the expression of these genes.

Cell division protein FtsZ was the target of nodule-specific cysteine-rich (NCR) peptides, and the interaction between them caused cell elongation (47, 48). Zhao et al. (49) found the importance of FtsZ2 protein in bacteroid differentiation. The higher expression of *ftsZ* in peanut bacteroids than in *S. flavescens* bacteroids might be another determinant factor for the formation of swollen bacteroids. Lytic transglycosylase domain (RL4716) was required for proper cell envelope function in *R. leguminosarum*, while the mutation of RL4716 domain influenced only the shape of free-living cells (50), and the differentiation of RL4716 mutant bacteroid in pea nodule was not checked.

The BacA protein is an ABC transporter, first identified in *S. meliloti* (51) as being crucial for bacteroid differentiation in alfalfa. BclA, homologous to BacA, was found in symbiotic *Bradyrhizobium* species associated with *Aeschynomene* plants and was essential for bacterial differentiation (52, 53), while *B. diazoefficiens* USDA110 does not require the *bclA* gene for BNF with *Aeschynomene afraspera* (53). In the *Mesorhizobium huakuii-Astragalus sinicus* symbiosis, protein BacA was indispensable for successful establishment of the symbiosis (54), similar to that in *Sinorhizobium meliloti* (51, 55). On the contrary, *bacA* gene homolog mlr7400 in *M. loti* MAFF303099 was dispensable for symbiosis with *Lotus japonicus* (56, 57). In strain CCBAU 051107 genome, a gene (WN72_20615) that had 60% protein homology to the gene encoding BclA was found. The expression of the WN72_20165 gene in the free-living cell was about 185.146 (FPKM values), while it decreased to 54.2544 in bacteroids of peanut. The function of BclA (WN72_20165) in determination of bacteroid development inside peanut nodule deserves to be further checked because the peanut nodule has many NCR peptides (58) which might function in bacteroid differentiation.

Peng et al. (59) compared the transcriptomes of nodulating and nonnodulating sister inbred peanut lines inoculated with the *Bradyrhizobium* strain and found some DEGs involved in symbiotic signaling interchange. Most recently, by comparison of peanut and other legume transcriptomes, Raul et al. (58) found Nod-factor-dependent epidermal crack entry and terminal bacteroid differentiation. Han et al. (60) and Wei et al. (61) determined alkaloids and flavonoids in *S. flavescens* by combining the metabolome and transcriptome, but they did not consider the effects of rhizobial nodulation on the synthesis of the medicinal ingredients. Our current studies may close the gap between these two kinds of studies and promote the elucidation of the pathway involved in alkaloid biosynthesis in *S. flavescens* (6, 60–62). In the future, dual-transcriptome analyses combining both symbiosis partners are needed to explore the interaction and signal exchange between them.

## MATERIALS AND METHODS

**Bacterial strain and plant growth.** Strain CCBAU 051107 of *B. arachidis* was isolated from effective root nodules of peanut (1) and *S. flavescens* (9, 12). The varieties of peanut and *S. flavescens* used in the present study were Yuhua no. 15 and Zhenku no. 2, respectively. Liquid bacterial cultures were prepared by obtaining a loopful of bacteria from YEMA (yeast extract-mannitol agar) slants (63) and inoculating 5 mL of yeast extract-mannitol (64). The liquid cultures were incubated at 28°C on a rotary shaker at 160 rpm. Water-agar (0.8%) was used to support the pregermination of plant seeds at 28°C for 2 to 3 days, after surface sterilization using 2.5 to 3.0% (wt/vol) sodium hypochlorite for 5 min as described previously (9, 10). After being germinated, seedlings were transplanted into sterilized vermiculite soaked with nitrogen-deficient nutrient solution (63) for growth, and each seedling was inoculated with 1 mL of the logarithmic growth phase of rhizobial strain CCBAU 051107 (diluted to an optical density at 600 nm [$OD_{600}$] of 0.2), while plants without inoculation were included as controls. All of the plants were grown in a greenhouse with an automatically controlled light/dark cycle of 12 h/12 h and harvested about 45 days postinoculation (dpi) to check the nodulation. Nitrogenase activity of root nodules was determined by the widely applicable acetylene reduction assay (ARA) with gas chromatography (65, 66) for each plant. Meanwhile, root nodules were collected for microscopic observation and RNA extraction according to the following procedures.

**Collection of free-living bacterial cells used for RNA extraction.** After activation on YEMA medium, the rhizobial strain was transferred into TY (tryptone-yeast extract) liquid broth and cultured at 28°C with shaking (160 rpm) to an $OD_{600}$ of 4.5 to 5.5. The cultures were collected by centrifugation at $1,000 \times g$ for 15 to 20 min. The collected cells were used to extract total RNA immediately or were frozen by adding liquid nitrogen into the centrifuge tube and then stored at −80°C for subsequent RNA extraction.

**Collection of root nodules used for RNA extraction.** To prevent the degradation of target RNA, the collected root nodules from peanut and *S. flavescens* plants were immediately frozen in a 50-mL centrifuge tube with liquid nitrogen during the collection process and then transferred immediately to −80°C for further storage.

**Reagents and kit used in RNA extraction, PCR, and sequencing.** The total RNA extraction kit RNAiso Plus (catalog no. 9108) was purchased from TaKaRa Bio, Inc. High-salt precipitation solution was purchased from Beijing Liuhetong company. Diethyl pyrocarbonate (DEPC)-treated pure water (catalog no. BN24052) was purchased from Beijing BioDee Biotechnology Co., Ltd. A quantitative PCR kit StarScript II first-strand cDNA synthesis mixture (catalog no. A223) and reverse transcription kit StarScript II first-strand cDNA synthesis mixture were purchased from GenStar. PCR primer synthesis and high-throughput sequencing of RNA samples were carried out at the Beijing Huada Gene Company.

**Extraction of RNA from rhizobia and bacteroids.** Separation and purification of bacteroids from root nodules of *S. flavescens* and peanut were performed according to the method of Day et al. (67), using the reagents polyvinylpyrrolidone (PVP) and Percoll. The bacteroids were used for subsequent RNA extraction. About 0.1 g rhizobial or bacteroid cells was added into a mortar precooled with liquid nitrogen and thoroughly ground into powder. The powder was transferred into RNase-free tube containing 1 mL RNAiso Plus reagent (TaKaRa, catalog no. 9108) and mixed thoroughly. The RNA extraction was followed by the manufacturer's instructions of TaKaRa. The purity of the RNA extract was detected using Agilent 2100 Bioanalysis machine: the pure sample used for further analyses should present the values of $OD_{260}/OD_{280}$ values of 1.8 to 2.0, 23S rRNA/16S rRNA values of ≥1, and an RNA integrity number (RIN) of ≥8.

**Preparation of semithin section of root nodules for microscopic observation.** The embedding of the root nodules in resin was carried out using Technovit 7100 reagent kits (Kulzer, Germany). The root nodules of peanut and *S. flavescens* were cut into two parts and fixed in 2.5% (vol/vol) glutaraldehyde solution. The preparation of nodule slices used for light and electron microscopes followed methods described previously (9, 68).

**Construction of cDNA library.** After being qualified, the above-mentioned total RNA samples were used for subsequent operations. RNA-free DNase I was added to the RNA samples and were incubated at 37°C for 30 min to digest DNA. rRNAs from plants and bacteroids were removed by using RiboMinus (Thermo Fisher Scientific). The plant mRNAs were enriched and removed by using oligo(dT) magnetic beads, leaving the remaining bacteroids' mRNA. The cDNA library was constructed and sequenced by the Illumina HiSeq 2500 platform. The reading length was 100 bp, and the cDNAs were sequenced by paired ends. The raw data obtained after sequencing was raw reads. After removal of the reads with adapters, the sequences containing more than 5% N bases, and low-quality reads, the data obtained from the company were called as clean reads.

**Analysis of RNA sequences.** Using TopHat software (based on Bowtie2 aligning tool), the data of the RNA transcriptome were matched to the reference genome and gene sequences of *B. arachidis* CCBAU 051107 (NCBI accession no. NZ_CP030050.1) with less than 5 mismatched bases. The Cufflinks software was used to assemble the transcripts, estimate the abundance of transcripts, and detect the differences in expression and variable expression shared among samples. When two samples were selected for finding the differentially expressed genes (DEGs), the standard was that the absolute value of $\log_2$ FC was ≥1 and the $P/q$ value was ≤0.05. The $\log_2$ FC represented the multiple of gene expression, and the general absolute value of ≥1 represented the difference. The $P$ value represented the degree of gene difference, and the standards were <0.05, <0.01, and <0.001, meaning significant, very significant and extremely significant difference, respectively. The $q$ value was a corrected $P$ value, and it was stricter than the $P$ value.

Statistical analyses of the common and specific expression genes ($\log_2$ FC of >2 or <−2) in bacteroids of peanut and *S. flavescens* were performed to draw a Venn diagram using the online tool Venny v.2.1.0 (69), which was manually adjusted and colored in Microsoft PowerPoint. DEGs were also evaluated based on the COG (Clusters of Orthologous Groups) protein database (70) among the bacteroids from two different host plants versus the free-living rhizobial cells. The KEGG (Kyoto Encyclopedia of Genes and Genomes) pathway database (71) was used to analyze the upregulated and downregulated genes in the metabolic pathways. The top 30 upregulated DEGs and downregulated DEGs were chosen for detailed comparison according to their importance in symbiosis formation, such as genes evolved in nodulation, nitrogen fixation, and lipopolysaccharide (LPS) synthesis (*lpx*), which play a special role in promoting the establishment of symbiosis between legumes and rhizobia (72).

**qRT-PCR.** Fluorescence real-time quantitative PCR (qRT-PCR) was performed to evaluate the reliability of RNA-Seq transcriptome results. By comparing with the gene expression in free-living rhizobial cells, a total of 14 genes, including genes upregulated or downregulated simultaneously in two kinds of bacteroids (swollen bacteroids formed in peanut nodules and nonswollen bacteroids formed in *S. flavescens*) or genes upregulated or downregulated only in one kind of bacteroid, were selected for qRT-PCR analysis. RNAs were extract by the method described above. The StarScript II first-strand cDNA synthesis mix reverse transcription kit (from GenStar) was used to reverse the RNA to cDNA according to the manufacturer's protocol. qRT-PCR was performed using 2× RealStar green power mixture (GenStar) by following the manufacturer's protocol. The 16S rRNA gene of strain CCBAU 051107 was used as an internal reference for quantitative analysis of relative expression.

**Data availability.** The RNA-Seq data have been submitted to the SRA database accession no. PRJNA819203.

## ACKNOWLEDGMENTS

This work was supported by funding from the NSFC (no. 31770039) to W. F. Chen.

We thank M. Li and Mnasri Bacem for their improvement of the manuscript.

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
