## [Reviewer comments · Microbiology Spectrum]

Microbiology Spectrum

Bacteroid development, transcriptome and symbiotic nitrogen-fixing comparison of *Bradyrhizobium arachidis* in nodules of peanut (*Arachis hypogaea*) and medicinal legume *Sophora flavescens*

Wenfeng Chen, Xiang Fei Meng, Yinshan Jiao, Chang-Fu Tian, Xinhua Sui, Jian Jiao, Entao Wang, and Sheng Jun Ma

Corresponding Author(s): Wenfeng Chen, China Agricultural University

Review Timeline:

Submission Date:	March 24, 2022
Editorial Decision:	July 5, 2022
Revision Received:	September 8, 2022
Editorial Decision:	October 3, 2022
Revision Received:	November 14, 2022
Accepted:	December 29, 2022

Editor: Jannell Bazurto

Reviewer(s): Disclosure of reviewer identity is with reference to reviewer comments included in decision letter(s). The following individuals involved in review of your submission have agreed to reveal their identity: Muhammad Asif Mansoor (Reviewer #1)

Transaction Report:

DOI: <https://doi.org/10.1128/spectrum.01079-22>

July 5, 2022

Dr. Wenfeng Chen
China Agricultural University
Yuanmingyuan Xilu No. 2
Beijing
China

Re: Spectrum01079-22 (Bacteroid development, transcriptome and symbiotic nitrogen-fixing comparison of *Bradyrhizobium arachidis* in nodules of *Arachis hypogaea* and *Sophora flavescens*)

Dear Dr. Wenfeng Chen:

Thank you for submitting your manuscript to Microbiology Spectrum. While the reviewers agree that the manuscript describes a study that was well executed and scientifically sound, it is overly observational in nature. More needs to be said that speaks to the conclusions drawn and overall biological message taken from this work.

Link Not Available

Sincerely,

Jannell Bazurto

Journals Department
Reviewer comments:

Reviewer #1 (Comments for the Author):

It is an outstanding study describing a unique field of knowledge making a behavioral comparison of bacteroids in two different plant types.

Here are some simple recommendations

1- Please improve the English grammar in text, many of the suggestions have been imparted in the review file, but further work

is needed.

2- Never start a sentence with an abbreviation or a number.

3- Avoid unnecessary information as recommended in the commented file, it makes text boring for a reader.

4- It's recommended to use the past tense form while describing the methodology and results, where applicable.

Reviewer #2 (Public repository details (Required)):

This study has done transcriptomics on nodule bacteroids of the two legumes peanut and *Sophora flavescens*. These are big data which need to be deposited at a public repository.

Reviewer #2 (Comments for the Author):

The paper is strong technically. The authors have included very detailed methodologies and were honest in their approach.

Staff Comments:

Preparing Revision Guidelines

Please return the manuscript within 60 days; if you cannot complete the modification within this time period, please contact me. If you do not wish to modify the manuscript and prefer to submit it to another journal, please notify me of your decision immediately so that the manuscript may be formally withdrawn from consideration by Microbiology Spectrum.

**Bacteroid development, transcriptome and symbiotic**
**nitrogen-fixing comparison of *Bradyrhizobium arachidis* in**
**nodules of *Arachis hypogaea* and *Sophora flavescens***

Wen Feng Chen^{a,*}, Xiang Fei Meng^a, Yin Shan Jiao^a, Chang Fu Tian^a, Xin Hua Sui^a,
Jian Jiao^a, En Tao Wang^b, Sheng Jun Ma^c

a. State Key Laboratory of Agrobiotechnology, Beijing 100193; College of
Biological Sciences and Rhizobium Research Center, China Agricultural
University, Beijing 100193, P. R. China.

b. Departamento de Microbiología, Escuela Nacional de Ciencias Biológicas,
Instituto Politécnico Nacional, México City, D. F. 11340, México.

c. College of Food Science and Pharmacy, Xinjiang Agricultural University, Urumqi
830052, Xinjiang Uygur Autonomous Region, China

*, Corresponding author: Wen Feng Chen, Email: chenwf@cau.edu.cn

**Abstract**

A rhizobial strain *Bradyrhizobium arachidis* CCBAU 051107 differentiates into
swollen bacteroids in nodules of peanut (*Arachis hypogaea*) and non-swollen
bacteroids in the extremely promiscuous medicinal legume *Sophora flavescens* were
compared on their morphologies, transcriptomes and nitrogen fixation efficiency of
their root nodules for the first time. Nitrogenase activity of peanut nodules was three
21 times higher than nodules of *S. flavescens* because of the different modifications of
22 symbiotic effectiveness by the host plants respectively. Transcriptome comparisons of
23 the two kinds of bacteroids showed that differentially expressed genes were involved
in the energy production and conversion, metabolism of carbohydrates, transport and
transcription of amino acids, peptidoglycan synthesis and cycles. For the two types of
bacteroids, the genes involved in biological nitrogen fixation and energy metabolism
were both up-regulated, while genes involved in DNA replication, bacterial
chemotaxis, flagellar assembly were both significantly down-regulated. There were
obviously different expressions of genes involved in the urea degradation and urease
activation, PHB metabolism and peptidoglycan biosynthesis between the two type
bacteroids. Our jobs revealed the differences of bacteroid metabolism of these two
host plants based on the RNA transcriptome levels. It will enhance our understanding
of the destiny, adaptive mechanism, and metabolic difference of the bacteroids in
different host legumes, and will be helpful for us to regulate the differentiation and
status of bacteroids to obtain higher nitrogen fixing activity when the rhizobium
establishes symbiosis with specific legumes, the important oil crop peanut and
medicinal legume *S. flavescens*.

**Key words**

*Bradyrhizobium arachidis*; *Arachis hypogaea*; Swollen bacteroid and non-swollen
bacteroid; *Sophora flavescens*; Differentiation; Root nodule; Legume; Rhizobia

**Introduction**

Peanut or groundnut (*Arachis hypogaea* L.) is an important annual oil crop
widely grown around the world and it can establish determinate root nodules for
symbiotic nitrogen fixation mainly with slow-growing rhizobia classified in the
species *Bradyrhizobium arachidis* (1), *Bradyrhizobium guangdongense*,
*Bradyrhizobium guangxiense* (2) and others (3, 4). Inside the root nodules of peanut,
the bradyrhizobia differentiate into spherical morphotype of bacteroids (5, 6).

The medicinal legume *Sophora flavescens* (7, 8) (common name “Kushen” in
Chinese that means “bitter ginseng”) is an extremely promiscuous host plant
establishing symbioses with more than 35 rhizobial species in the Classes of
*Alphaproteobacteria* and *Betaproteobacteria* (9-11). This plant can form
indeterminate root nodules with *Rhizobium leguminosarum* (bv. *trifolii* and bv. *viciae*),
*Sinorhizobium meliloti*, *Mesorhizobium loti*, *Rhizobium etli* bv. *phaseoli*, and the
currently studied peanut microsymbiont *Bradyrhizobium arachidis* (9, 12), despite the
distinct phylogenetic positions of species and symbiovars of them.

Because of the promiscuously symbiotic properties of the host *S. flavescens*, it
can be used as an excellent plant model to study the development and physiology of
various rhizobia and bacteroids inside its root nodules (11). Our previous study ever
compared physiologically the phosphate starvation of rhizobia without terminal
differentiation in various legume nodules and found in that the strain *R.*
*leguminosarum* bv. *viciae* Rlv3841 underwent characteristically nonterminal and
terminal differentiations in nodules of *S. flavescens* and pea, respectively (11). In
another studies, we ever observed elliptical or sphere-like bacteroids and many
rod-shape bacteroids inside root nodules of *S. flavescens* infected by strain
*Mesorhizobium loti* NZP 2213 (9). Furthermore, bacteroids of NZP 2213 in *S.*
*flavescens* nodules showed more specific shapes including branching morphology, a
characteristic property of swollen or terminal differentiation bacteroids (our

unpublished data), according to the classification of Oono et al. (13) for bacteroids.
Therefore, *S. flavescens* is a so interested host plant for studying the development of
different rhizobia inside its root nodules.

Going back to the above mentioned *Bradyrhizobium arachidis*, the strain
CCBAU 051107 of this species could induce root nodules on both hosts, determinate
nodules on peanut (9) and indeterminate nodules on *S. flavescens* (12). So it could be
used for comparative studying about the differentiation and metabolism of
microsymbionts in different kinds of root nodules. As few studies have ever focused
on the metabolism and gene expression of one rhizobial strain in different host root
nodules, especially for the bacteroids inside of important oil crop peanut nodules, the
aim of present study was to reveal the differences in the physiology and
nitrogen-fixing efficiency of this strain associated with these two hosts. In addition, as
yet, no study has been carried out to investigate the transcriptome of bacteroids inside
the root nodules of peanut and *S. flavescens*. Therefore, the development, physiology
and metabolism of the bacteroids in different hosts need to be elucidated and the
results will be interesting in the field of symbiotic interaction between rhizobia and
legumes.

**Materials and methods**

**Bacterial strain and plant seedling growth**

Kept in our laboratory, *B. arachidis* strain CCBAU 051107 is a slow-growing
bacterial strain originally isolated from root nodule of peanut (1), and it was also an
effective microsymbiont for *S. flavescens* (9, 12), which could promote the growth
and increase of active ingredient of this medicinal legume (12). Varieties for peanut
and *S. flavescens* were Yuhua No. 15 and Zhenku No. 2, respectively. Yeast
extract-Mannitol (YM) (14) agar or Tryptone-Yeast extract (TY) (15) broth were used
to culture the rhizobial strain at 28 °C. Water-agar (0.8%, g/w) was used to support the

germination (28 °C, two to three days) of the plant seeds after sterilization using
2.5-3.0% (w/v) sodium hypochlorite (NaClO) for 5 min. Seeds of *S. flavescens* were
treated using concentrated sulfuric acid for 20 min previous to the surface-sterilization
to remove the waxy layer of seed coat. Sterilized deionized water was used to rinse
the surface sterilized seeds for 6-7 times for removing the disinfectant. Sterilized
vermiculite soaked with a low-nitrogen level solution and trace element (14) were
used to support the growth of the germinated seedlings and each seedling was
inoculated with 1 mL logarithmic growth phase of rhizobial strain CCBAU 051107
(diluted with sterilized 0.8% normal saline to $OD_{600} = 0.2$). Growth of the plants was
kept in greenhouse automatically controlled cycle of 12 h light /12 h dark. The plants
without inoculation were used as controls. The plants were harvested about 45 days
after inoculation, and the nodulation of the plant roots was checked. Nitrogenase
activity of the root nodules per plant was determined by widely applicable acetylene
reduction assay (ARA) with gas chromatography (16, 17).

**Collection of free-living bacterial cells used for RNA extraction**

After activation on YM agar medium, the rhizobial strain was transferred into TY
liquid broth and cultured at 28 °C with shaking (160 rpm) to density of 4.5-5.5
(OD_{600nm}). The cultures were collected by centrifugation (4000 rpm, 15-20 min). The
collected cells were used to extract the RNA immediately, or the liquid nitrogen was
added into the centrifuge tube and then stored in a refrigerator at -80 °C for
subsequent RNA extraction.

**Collection of root nodules used for RNA extraction**

To prevent the degradation of target RNA, the collected nodules of peanut and *S.*
*flavescens* should be immediately transferred into a 50 ml centrifuge tube filled with
liquid nitrogen, and the nodules should always be kept in the liquid nitrogen
environment during the collection process. After collecting the nodules, the centrifuge

tubes were transferred immediately into a refrigerator at -80 °C.

**Reagents and kit used in RNA extraction, PCR and sequencing**

The following reagents were used to extract RNA and for PCR. Total RNA extraction
Kit RNAiso Plus (Cat. #9108) was purchased from Takara Bio. Inc. High salt
precipitation solution was purchased from Beijing Liuhetong company. DEPC pure
water (#BN24052) was purchased from Beijing BioDee Biotechnology Co., Ltd.
Quantitative PCR kit StarScript II First strand cDNA synthesis mix (#A223) and
reverse transcription kit StarScript II first strand cDNA Synthesis Mix were purchased
from GenStar company. PCR primer synthesis and sequencing, and high-throughput
sequencing were carried out in Beijing Huada Gene Company.

**Extraction of RNA from rhizobia**

About 0.1 g rhizobial cells were added into mortar pre-cooled by liquid nitrogen and
ground into powder thoroughly. Then, the powder was transferred into RNase-free
tube containing 1 mL RNAiso Plus reagent (Takara, #9108) and mixed thoroughly.
After being kept for 5-10 min at room temperature, the mixture was centrifuged at
4 °C for 15 min (12,000 rpm). The supernatant was transferred to a clean 1.5 mL
centrifuge tube and was mixed with 2 ml chloroform. After another 5-10 min
incubation at room temperature, it was centrifuged again at the same condition. The
supernatant (about 400 µL) was transferred into another new 1.5 mL centrifuge tube.
One volume of isopropanol and one volume of high salt precipitation solution were
added to the supernatant phase. The solution was gently reversed, mixed, and
incubated at room temperature for 10 minutes, and then incubated at -20 °C overnight.
After centrifugation at 4 °C for 10 min at 12,000 rpm, the supernatant was removed
without touching the precipitate. One mL of 70% ethanol was added and centrifuged
at 4 °C for 10 min at 12,000 RMP. After discarding the supernatant, 1 mL of absolute
ethanol was added to each tube for storage and transportation. When the RNA was

ready to use, the tube was centrifuged, the ethanol phase was discarded, and the tube
was dried a minute at room temperature, then small amount of DEPE pure water
(about 50 μ L) was added to dissolve RNA. The purity of the RNA extract was
detected using Agilent 2100 Bioanalysis machine. The values should be OD_{260}/OD_{280}
= 1.8-2.0, 23S rRNA:16S rRNA ≥ 1 , RNA integrity number (RIN) ≥ 8 and then the
RNA can be used for further analyses.

**Preparation of semithin section of root nodules for light microscopic observation**

The embedding of the root nodules in resin was carried out using Technovit[®] 7100
reagent kits (Kulzer, Germany). The root nodules of peanut and *S. flavescens* were
respectively cut into two parts and fixed in 2.5% glutaraldehyde solution. The fixed
root nodule tissues were soaked into 0.05 M cacodylate buffer for 1 h and then
transferred into sterilized deionized water for 30 minutes. The nodule tissues were
subsequently immersed in 30%, 50%, 70%, 95% and 100% ethanol for 2 h
respectively. Then they were immersed in ethanol/Technovit[®] 7100 base solution (1:1,
v/v) for 12 h. Move the nodule tissues to Technovit[®] 7100 base plus Hardener I (1%)
solution for 24 h. Transfer the nodule tissue to mixture of Technovit[®] 7100 base plus
Hardener I (1%) + Hardener II (1.5mL) for 23 h and leave it in a 37 °C incubator for 1
164 h. The root nodule tissue was embedded and cut into 2 μ m semithin sections using
Leica C6i slicer, then were dyed with toluidine blue dye for 40 s, then rinsed with
sterilized deionized water, and then dried in air. The observation of the dyed thin slice
was carried using light microscope.

**TEM observation of root nodules of *S. flavescens* and peanut**

The nodule was cut longitudinally for *S. flavescens* rod-shaped nodule and equally
divided from the equatorial line for the spherical peanut nodule and then washed with
0.1M PBS buffer, fixed with 1% citric acid, and then rinsed with 0.1M PBS buffer.
Then it was dehydrated subsequently with 30%, 50%, 70%, 80%, 90% and 100%
acetone. After that, the root nodule was embedded in Spurr's resin and sectioned with

Leica C6i microtome to 80 nm ultra-thin slices. The slices were stained with uranyl
acetate and lead citrate. Finally, JEM-1230 transmission electron microscope (JEOL
USA Inc.) was used to observe the sections and photos were taken.

**Enrichment of bacteroids**

Extraction of bacteroids from root nodules was followed by the methods described
previously (18) with some following revisions. Three grams of root nodules of *S.*
*flavescens* or peanut respectively were taken out from -80 °C refrigerator and were
added into a mortar pre-cooled with ice. A total 15 mL of pre-cooled 1% PVP
extraction buffer (including 10 mM DTT, 300 mM sucrose solution, 10 mM pH 7.0
phosphoric acid buffer, 2 mM MgCl₂) and a small amount of quartz sand was added
and the nodules were ground evenly. The mixture was transferred into a new 50 mL
centrifugal tube and centrifuged at low-speed (400 g), 4 °C for 10 min and transferred
into another new 50 mL centrifuge tube. An aliquot of 20 mL of extraction buffer was
added to the pellet to re-extract it once, then centrifuged at 400 g, 4 °C for 10 min,
and the supernatant was transferred to above 50 ml centrifuge tube. After
centrifugation at 10,000 g for 10 min, the supernatant was discarded, and the pellet of
bacteroids was resuspended in 10 mL of extraction buffer in a 40 mL centrifuge tube
(especially used for horizontal centrifuge), and then add 10 mL 30% Percoll[®], 5 mL
60% Percoll[®] and 5 mL 80% Percoll[®] in turn to the bottom of the centrifuge tube.
After centrifugation at 4,000 ×g in a horizontal centrifuge at 4 °C for 15 min, the
liquid is stratified, and a layer of white pad-like precipitate between 60-80% Percoll[®]
layer was bacteroids. The layer containing bacteroids on the upper edge of 80%
Percoll[®] was extracted by using pipette tip cut off the tip end, then diluted with
normal saline and centrifuge at 10,000 g for 5 min to discard the supernatant. The
precipitates were now the expected bacteroids.

**Construction of cDNA library**

After being qualified, the above-mentioned total RNA samples were used for
subsequent operations. RNA-free DNase I was added to the RNA samples and they

were incubated at 37 °C for 30 min to digest DNA. Ribosomal RNAs (rRNA) from
plants and bacteroids were removed by using RiboMinus™ (Thermo Fisher
Scientific). The plant message RNA (mRNA) were enriched and removed by using
Oligo dT magnetic beads, the remaining were bacteroids' mRNA. The mRNA was
segmented using Fragmentation buffer, and then transcribed into cDNA by random
primers and reverse transcriptase. The DNA was repaired at the end, and then the
poly(A) bases were added to the end and connected with the sequencing adapters. The
products were purified and about 200 bp fragments were selected as templates for
PCR amplification. The constructed cDNA library was sequenced by Illumina
HiSeq™ 2500 platform. The reading length was 100 bp and they were sequenced by
paired ends. The raw data obtained after sequencing was raw reads and it should
remove the reads with adapters, the sequences containing more than 5% N bases, and
low-quality reads. The data obtained from the company were called clean reads.

**RNA-seq**

By using TopHat software (based on Bowtie2 aligning tool), the data of RNA
transcriptome were matched to the reference genome and gene sequences of *B.*
*arachidis* CCBAU 051107 (NCBI accession no. NZ_CP030050.1) with less than 5
mismatched bases. The Cufflinks software was used to assemble the transcripts,
estimate the abundance of transcripts, and detect the differences of expression and
variable sheared among samples. When two samples were selected for finding the
differential expressed genes (DEGs), the standard was that the absolute value of log₂
(fold change) was ≥ 1 and p/q value was ≤ 0.05 . Log₂ (fold change) represents the
multiple of gene expression, and the general absolute value ≥ 1 represents the
difference. The p value represents the degree of gene difference, and the standard is
less than 0.05, 0.01, and 0.001, meaning significant difference, very significant and
extremely significant, respectively. The q value is a corrected value to p , and it is
stricter than p value.

**qRT-PCR**

Fluorescence real-time quantitative PCR (qRT-PCR) was used to evaluate the
reliability of RNA-seq transcriptome results. By comparing with the gene expression
in free-living rhizobium, a total of 15 genes, including genes up-regulated or down
regulated simultaneously in two kinds of bacteroids, or genes up-regulated or down
regulated only in one kind of bacteroid, were selected to perform the qRT-PCR. RNA
extraction was followed the method described above. StarScript II first strand cDNA
synthesis mix reverse transcription kit (from GenStar[®]) was used to reverse the RNA
to cDNA. The primers used in qRT-PCR were shown in Supplementary Table S1.
qRT-PCR was performed using 2 × RealStar green power mixture (GenStar[®]) and
followed by the manufacture protocol. The 16S rRNA gene of strain CCBAU 051107
was used as internal reference for quantitative analysis of relative expression.

**Data availability.**

The RNA-Seq data have been submitted to the SRA databases under accession
number [PRJNA819203](https://www.ncbi.nlm.nih.gov/sra/PRJNA819203).

**Results**

**Comparison of morphology and section of peanut and *S. flavescens* root nodules**

The *B. arachidis* strain CCBAU 051107 induces determinate root nodules on peanut
(*A. hypogaea*) (Fig. 1 A1, A2) and indeterminate nodules on *S. flavescens* (Fig. 1 B1,
B2), respectively (Fig. 1). The spherical peanut nodules displayed dark red section
bearing higher content of leghemoglobin (Fig. 1 A1, A2). The rod-shaped root
nodules of *S. flavescens* presented light red section (Fig. 1 B1 and B2). The total
numbers of root nodules per plant of peanut were distinctly higher than those of *S.*
*flavescens* (Fig. 1 A1 and B1). The average diameter of mature peanut root nodules
was about 1 mm (Fig. 1 A2), while the size of root nodules of *S. flavescens* was 2 mm

in length by 1 mm in width (Fig. 1 B2).

**Microstructure and bacteroids of peanut and *S. flavescens* nodules**

Nodule sections and bacteroids inside the nodules of these two host plants infected by
strain CCBAU 051107 were observed under general light microscope and
transmission electron microscope (TEM) (Fig. 1 A3, A4, B3 and B4). The root nodule
of peanut has spherical meristem zone (MZ) and is classical determinate type (Fig. 1
A3), while *S. flavescens* is indeterminate type with apical meristem zone (MZ) (Fig. 1
B3) on the apex of the nodule. The area of nitrogen fixation (NF) zone of peanut
nodule (Fig. 1 A3) is smaller than that of *S. flavescens* nodule (Fig. 1 B3),
corresponding to the red color ranges in their responding nodule sections (Fig. 1 A2,
B2), while the former has higher and homogeneous density of bacteroids (Fig. 1A3,
few white plaque) than the latter (Fig. 1B3, many white plaques). Bacteroids inside
peanut root nodule is sphere with polyphosphate (PP) inclusions and has no obvious
poly- β -hydroxybutyrate (PHB) particles (Fig. 1 A4), while bacteroids inside nodule of
*S. flavescens* are rod and the length of them is $< 4 \mu\text{m}$ (Fig. 1 B4). In addition, an
obvious accumulation of PHB particles was found in bacteroids of *S. flavescens*
nodule (Fig. 1 B4). Only one spherical bacteroid was enclosed in each symbiosome in
peanut nodule cell with no peribacteroid space (PBS), while many bacteroids (up to
more than 10) were housed in a symbiosome of *S. flavescens* nodule cell with distinct
PBS (Fig. 1 B4).

**Activity of ARA of root nodules**

Nitrogenase activity of the root nodules was determined using acetylene reduction
assay (ARA) method and the result represented by production of ethylene (C_2H_4)
from reduction of acetylene (C_2H_2) was shown in Fig. 1C. The activity of
nitrogenase of peanut root nodules was $12.06 \pm 2.04 \text{ nmol/h/mg}$, which was about 3
279 times higher than that of *S. flavescens* ($3.81 \pm 0.58 \text{ nmol/h/mg}$).

Sequencing data and mapping to known genome sequence

The RIN values of all RNA samples were above 7.5. All RNA samples met the
high-quality requirements for RNA-seq database construction and they were sent to
the company to build the database and subsequent sequencing.

The sequence data were used to map the reference sequence for *B. arachidis* CCBAU
051107 genome (accession no. NZ_CP030050 in NCBI database). Total clean reads
of each sample were more than 10 Mbyte (MB), and the read length of double
terminal sequencing is 100 bp, so the total base number of each sample is more than 1
Gigabyte (GB). The mapping ratios of total clean reads on the genome sequence of *B.*
*arachidis* CCBAU 051107 were more than 90% (free-living bacterium), 28% (peanut
bacteroid) and 24% (*S. flavescens* bacteroid), respectively. The sequencing depth of
bacteroids (200Mb) also reached more than 30 times of the rhizobial genome size
(7MB), which fully met the requirements.

Comparison of gene expression and verification of transcriptional data by 294 qRT-PCR

A total 15 genes were selected for qRT-PCR verification of the RNA-Seq results.
Results (Fig. 1 D) showed that there was a good consistency between the RNA-Seq
data and qRT-PCR data with Pearson correlation coefficient of 0.853 ($P < 0.0001$) and
liner $R^2 = 0.727$.

Results from comparison of differentially expressed genes (DEGs) showed that total
1046 DEGs were found in comparison between peanut bacteroids and free-living
bacteria, including 482 up-regulated genes and 564 down-regulated genes (Fig. 2A).
Total 1590 DEGs were found in comparison between bacteroids of *S. flavescens* and
free-living rhizobia, including 934 up-regulated genes and 656 down-regulated genes
(Fig. 2B). Of these up-regulated genes encoding for components of nitrogenase,
*nifHDK*, rank first top in both kinds of bacteroids. Distinct DEGs of known *ureABEF*,

involved in urease were highlighted in bacteroids of *S. flavescens* (Fig. 2B). The
expressions of *ureABE* genes in bacteroids of peanut are not so high, with log₂ Fold
Change values of 3.87, 2.82 and 2.52, respectively, compared with that of free-living
bacteria (Fig. 2A). In addition, from the volcano plot in Fig. 2, it is clear that there are
two “wings” on the top of the volcanic vent, each representing the top up- and
down-regulated expression genes in each kind of bacteroids of these two host plants
(Fig. 2).

**COG analysis of differential gene results**

Transcriptome differential genes (DEGs) were also evaluated based on the protein
database of clusters of orthologous groups (COG) among the bacteroids from two
different host plants vs. the free-living rhizobia (Fig. 3). Results from Fig. 3 showed
that the significantly changed DEGs either up-regulated or down-regulated were
mainly involved in amino acid transport and metabolism, energy production and
conversion, carbohydrate transport and metabolism, transcription, inorganic transport
and metabolism, besides the unknown function and predicted general function genes.
The down-regulated genes mainly participated in cellular processes and signaling, cell
motility, DNA replication, cell division and chromosome, and the related genes of
translation, ribosomal structure and biogenesis. The trends of these changed DEGs in
two kinds of bacteroids were generally consistent. The total numbers of DEGs in *S.*
*flavescens* bacteroids (Fig. 3A) were twice higher than those in peanut bacteroids (Fig.
3B) compared to the free-living rhizobia, especially in the COG category of amino
acid transport and metabolism.

**Differentially expressed genes matched in KEGG pathway database**

KEGG pathway database can be used to analyze the up-regulated and down-regulated
genes in the metabolic pathways, such as cellular surface structure of rhizobia (*exo*,
*lpx*), nodulation (*nod*, *nol* and *noe*), nitrogen-fixing (*nif*, *fix*) and others (Fig. 4 and Fig.

5).

The first 30 up-regulated and down-regulated expression genes in bacteroids of *A.*
*hypogaea* and *S. flavescens* were compared respectively with the free-living bacteria
(Fig. 4A and 4B). Clearly, most of the up-regulated genes (*fix*, *nif*) were involved in
nitrogen fixation and nitrogen metabolism, and of these 30 up-regulated genes, total
12 genes were common in expression profiles of both kinds of the bacteroids. In
addition, the genes involved in urea degradation (*ureA*, *ureB*) and urease activation
(*ureF*) were highly up-regulated in bacteroids of *S. flavescens* (ranked in No. 1 and 3
for *ureA* and *ureB*, respectively) but they were not expressed significantly in peanut
bacteroids (Fig. 4A and 4B). On the contrary, of these 30 down-regulated expression
genes, total 10 genes were common in both kinds of the bacteroids and most of them
(7) were hypothetical proteins (HP) (Fig. 4).

**(1) Expression of taxis, motivity and flagellin related genes**

The expression of genes involved in chemotaxis, motivity and flagellin were
down-regulated in both of the two kinds of bacteroids (Fig. 5A), such as genes of *mcp*
(for membrane sensing protein), *cheA*, *cheR*, *cheZ* and *motA*. In addition, expression
of these genes in peanut bacteroids was much down-regulated than those in bacteroids
of *S. flavescens*.

**(2) Expression of genes related to the structure of the surface of rhizobia**

Exopolysaccharides (EPS) of cellular surface structure is synthesized by *exo* gene
cluster. The values for represented *exo* genes are not very consistent among the
free-living rhizobia and the two kinds of bacteroids. In general, the *exo* gene
expression in these three kinds of bacteria and bacteroids were lower or slightly
down-regulated (Fig. 5B). Except for the up-regulated expression of *exoZ* gene, and
the down-regulated expression of *exoT* and *exoU* genes, other gene expression

changes are not significantly in bacteroids of peanut. In contrast, in bacteroids of *S.*
*flavescens*, the two genes *exoM* and *exoQ* are up-regulated significantly, but the *exoZ*
gene is down-regulated significantly, and the changes of other genes are not
significant.

Genes evolved in lipopolysaccharide (LPS) synthesis (*lpx*) and the products of LPS
have special role in promoting the establishment of symbiosis between rhizobia
legumes and rhizobia in the late stage of rhizobia development (Frayssé et al., 2002).
It can be seen from Fig. 5B that the expression of *lpxK* was up-regulated in bacteroids
of peanut and *lpxB* was up-regulated in bacteroids of *S. flavescens*. While the
expression of other *lpx* genes in the two kinds of bacteroids are similar to those in the
free-living rhizobia. Likewise, the changes in expression level of these *lpx* genes was
not absolutely significant and is similar the level of *exo* genes (Fig. 5B).

**(3) Expression of nitrogen fixation genes**

The expression of *nif* and *fix* genes in symbiotic nitrogen fixation process were
significantly increased both in the bacteroids of peanut and *S. flavescens* (Fig. 5C).
The expression of structure genes for nitrogenase (*nifHDK*) rank the first three tops
among the nitrogen fixation related genes. While the expression level of regulation
gene *nifA* is lower. In addition, the expression of most *nif* and *fix* genes in bacteroids
of peanut is higher than those of *S. flavescens*.

**(4) Expression of genes related to nodulation factor synthesis**

Nodulation genes mainly include *nod*, *nol* and *noe*. It can be seen from Fig. 5D that
the expression of *nod* gene (except *nodM* and *nodJ*) in two kinds of bacteria is not
significantly different from that in the free-living rhizobia, while the expression of
*nolABOUY* gene in the two kinds of bacteroids is significantly down-regulated
compared with those in the free-living rhizobia. The expression of *noeI* gene was

slightly up-regulated in peanut bacteroids. There was no significant difference in the
expression of structure genes for nodulation factor synthesis (*nodABC*) between the
two kinds of bacteroids, while the expression of regulation genes (*nodD1* and *nodD2*)
changed a lot especially for the *nodD2* gene that down-regulated significantly in
bacteroids of peanut.

**(5) Expression of TCA cycle related genes**

The *dctA* gene, responsible for the transport of tetracarboxylic acid to bacteroids as
carbon source, were highly up-regulated in both bacteroids (Fig. 5E), with FPKM
values ranged from 17 to 513 and 657, respectively in two bacteroids, while its
expression was lower in free-living rhizobia.

Most genes involved in TCA cycle related enzymes (Fig. 5E) were significantly
up-regulated in peanut bacteroids, including *ciz* (for citrate synthase), *mdh* (for malate
dehydrogenase), *mgo* (for malate:quinone oxidoreductase), *fumC* (for fumarase),
*sdhABD* (for succinate dehydrogenase), *sucDB* (for succinyl coenzyme). While the
expression of these above genes and *icdA* (for isocitrate dehydrogenase) gene were
significantly down-regulated in bacteroids of *S. flavescens* (Fig. 5E) compared with
that in bacteroids of *A. hypogaea*.

**(6) Expression of fatty acid metabolism related genes**

The genes expression that participated in the fatty acid metabolism in the bacteroids
of both hosts was not active or down-regulated (Fig. 5F).

**(7) Expression of PHB metabolism related genes**

The structural genes (*phbABC*) involved in PHB synthesis in bacteroids of peanut
were significantly down-regulated, while the expression of *phbD* gene (for PHB
depolymerase) increased compared to the free-living rhizobia; and the expression

level of *phbD* gene was a little bit higher than that of *S. flavescens* (Fig. 5G). No
significant differences were found in gene expression of *ascA2* (encoding for
acetyl-CoA synthetase) and *bdhA* (for 3-hydroxybutyrate dehydrogenase, central
enzymes to PHB degradation) of bacteroids of both host plants (Fig. 5G).

**(8) Expression of pentose phosphate pathway related genes**

The genes involved in pentose phosphate metabolism pathway were active and
up-regulated significantly in peanut bacteroids (Fig. 5H). Comparably, the expression
of these corresponding genes in bacteroids of *S. flavescens* was not so higher, even
down-regulated for the genes *tal* (for glucose-6-phosphate isomerase), *pfk* (for
6-phosphofructokinase), *pgl* (for 6-phosphogluconolactonase), *gnd* (for
6-phosphogluconate dehydrogenase) (Fig. 5H). While these genes were all
up-regulated expression in bacteroids of *A. hypogaea* (Fig. 5H).

**(9) Expression of cell division, peptidoglycan or murein cycle genes**

Expression of many genes involved in cell division, peptidoglycan or murein cycles in
free-living rhizobia and the bacteroids in two kinds of nodules are significantly
different (Fig. 5I). Significantly up-regulated expression genes ($q < 0.05$) in
bacteroids of *A. hypogaea* include genes encoding for FtsQ, FtsZ, MviN, MurJ, MltB
(lytic murein transglycosylase, WN72_12370), and Pbl (LysM peptidoglycan-binding
domain-containing protein, WN72_37580) (Fig. 5I). Significant down-regulated
expression genes ($q < 0.05$) in *A. hypogaea* bacteroids include genes encoding for
MurL (murein L, D-transpeptidase, WN72_03045), Pbp (peptidoglycan-binding
protein, WN72_09390, WN72_37390) and FlgJ (WN72_32400).

Comparatively, expression of many genes involved in cell division, peptidoglycan or
murein cycles (Fig. 5I) decreased in bacteroids of *S. flavescens* or changed not
significantly except the genes encoding for FtsW (WN72_33790), MltB (lytic murein

transglycosylase, WN72_06590, WN72_26565), Pbp (peptidoglycan-binding
protein, WN72_00345, WN72_06690), and Pbl (LysM peptidoglycan-binding
domain-containing protein, WN72_37580). Extremely significant differences ($q <$
0.001) between the bacteroids of the two hosts are the significantly up-regulated
genes of MviN (WN72_00145), FtsZ (WN72_37760) in peanut nodule, and the
significantly down-regulated genes of two Pbp (peptidoglycan-binding proteins,
WN72_09390, WN72_37390).

**(10) Expression of DNA replication and ribosome synthesis related genes**

It can be seen from Fig. 5J that the expression of DNA replication related proteins,
replication initiation enzyme (*dnaA*), single strand binding protein (*ssb*), DNA
helicase (*dnaB*) and DNA polymerase subunit genes (*dnaE* and *dnaQ*) are
down-regulated both in bacteroids of peanut and *S. flavescens*. The genes of large (*rpl*)
and small (*rps*) RNA subunits involved in ribosome syntheses were also
down-regulated both in bacteroids of peanut and *S. flavescens*.

**Discussion**

Previously, the influence of dual-host strain *Bradyrhizobium* sp. 32H1 on peanuts and
cowpeas was compared and the result showed that its swollen bacteroids in peanut
nodules conferred more net host benefit by two measures: return on nodule
construction cost (plant growth per gram nodule growth) and nitrogen fixation
efficiency (6). Our current results support the conclusion that swollen bacteroids in
peanut root nodules have higher gene expression (two to five times) than the
non-swollen bacteroids of *S. flavescens* related to nitrogen fixation (Fig. 1C and Fig.
5C). Differing from the previous studies, in root nodules of peanut and *S. flavescens*,
strain *B. arachidis* CCBAU 051107 differentiated into swollen and nonswollen types
of bacteroids, respectively, with different nitrogen fixation efficiency. The nitrogenase
activity of peanut nodules was about 3 times of that of *S. flavescens*, similar to the
results observed in peanut nodules and cowpea nodules that contained similar

contents of bacteroids (19). Though transcriptomic analyses of *Sinorhizobium* sp.
NGR234 bacteroids in determinate nodules of *Vigna unguiculate* and indeterminate
nodules of *Leucaena leucocephala*, were compared (20), this strain only develops into
non-swollen bacteroid and only one bacteroid was encapsulated by one peribacteroid
membrane in both kinds of nodules of the two host plants (20).

Through the comparison of COG functional categories (Fig. 3), the distribution of
DEGs in bacteroids was predominantly involved in the amino acid transport and
metabolism, consisting of the primary role of N₂-fixing bacteroids in both legume
nodules (21, 22). Our current results indicate that the functional classification of
differential genes (either up-regulated or down-regulated) is very similar in both
swollen and non-swollen bacteroids (Fig. 3). Five categories of DEGs (not shown in
Fig. 3, including extracellular structures, cytoskeleton, nuclear structure, chromatin
structure and dynamics) were not detected in both of the bacteroids, which were
similar to the previous results of Peng et al. (23), meaning that these processes may be
the characterization of eukaryotic cellular functions and are not appropriate in
descriptions of bacterial cells, especially for the bacteroids in legume root cells. The
reason may be due to the composites of the database of COG contain information
from eukaryotes.

Comparisons of various metabolic pathways by using KEGG (Fig. 5) confirmed the
differences between the two kinds of bacteroids. The expression of genes related to
function of chemotaxis, motivity and flagellin were down-regulated in both kinds of
bacteroids (Fig. 5A) compared with the free-living rhizobia, consisting with the
differentiation to specific bacteroids with undetectable amounts of gene expression
inside mature root nodules (24). The expression of gene WN72_32400 encoding for
flagellar assembly peptidoglycan hydrolase (FlgJ) (Fig. 5I) declined dramatically in
both kinds of bacteroids of peanut and *S. flavescens* compared with the free-living
motive rhizobia, further confirming the motionless status specializing in nitrogen

fixation inside the root nodules, since peptidoglycan-hydrolyzing activity of the FlgJ
protein, localized in periplasmic space of Gram-negative bacteria, is essential for
flagellar rod formation in *Salmonella typhimurium* (25, 26).

Surprisingly, *ureA* gene encoding for urease ranks No. 1 as the top 30 up-regulated
genes in bacteroids of *S. flavescens* (Fig. 4B). The other urease-related genes (*ureB*,
*ureF*) are also up-regulated in bacteroids of *S. flavescens*. While these *ure* genes were
not determined in the list of top 30 up-regulated in peanut bacteroids and or their
expressions were not high (Fig. 4A). Urease is one of the important enzymes in
nitrogen cycle (27), catalyzing the hydrolysis of urea to yield ammonia and carbamate
(28, 29). Urea can be utilized by bacterial symbionts in marine sponge *Xestospongia*
*testudinaria* (30). We estimate that the highest expression of *ure* genes in bacteroids
of *S. flavescens* might be related to the transfer of fixed N from bacteroids to the plant
root to synthesize the (oxy)matrine, active ingredients of *S. flavescens*, but this
estimation, as well as its impact on the syntheses of the (oxy)matrine content because
of nodulation of rhizobia to *S. flavescens* (31) and the requirement of nickel remain to
be studied later.

The significantly up-regulated gene expression related to TCA cycle and pentose
phosphate metabolism in peanut bacteroids than those of *S. flavescens* displayed the
difference in carbon and energy producing metabolisms between them (Fig. 5H). The
results were consistent with activity of genes in TCA cycle and pentose phosphate
metabolism further confirming the higher requirement for energy for nitrogen fixation
of swollen bacteroids (32). The genes involved in fatty acid metabolism in both kinds
of bacteroids are not active or down-regulated. It is speculated that the synthesis of
fatty acids is a process of energy consumption, which may compete requirement for
the nitrogen fixation of bacteroids. Therefore, fatty acid synthesis genes are not active
in bacteroids. Previous studies showed that the synthesis of fatty acids in bacteroids of
*B. diazoefficiens* USDA110 drastic declined (32). The bacteroid requirement for lipids

maybe as lower as its non-dividing nature or are supplied by host plant (32).

The PHB accumulation was found mainly in bacteroids of *S. flavescens* nodules but
few was observed in bacteroids of peanut nodules through the observation under TEM
(Fig. 1A4) and the result was consistent with the up- or down-regulated gene
expression of PHB syntheses (Fig. 5G), respectively, in these two kinds of bacteroids.
In addition, the PHB content was clearly not related with nitrogenase activity as
bacteroids in peanut nodules had few or no PHB, but the nodules had higher N₂-fixing
efficiency. Our current results demonstrated that PHB accumulation is variable
between rhizobial strains (33, 34) and host plants and the difference in nitrogen
metabolism of one strain in different root nodules needs to be explored further. In our
laboratory group, Wang (35) found that both strains CCBAU 45384 and CCBAU
53363 could infect peanut and differentiate into spherical bacteroids, while the
formerly contained amounts of PHB granules, but the latter did not (unpublished data).
Combining the results in our present study and in the study of Wang (35), PHB
accumulation is determined by different rhizobial strains and the host plants, and
these results are not consistent with other previous data stating that swelling
bacteroids accumulated little or no PHB (36). Our results also imply the possibility
that the PHB accumulation might decrease the efficiency of nitrogen fixation in
nodules of *Medicago truncatula*, as questioned by Wang *et al.* (37).

In previous studies, Li *et al.* (20) ever reported that the expression of many genes
involved in cell division was down-regulated in bacteroids of *Vigna unguiculata* and
*Leucaena leucocephala*. These results are consistent to the gene expression mode in
bacteroids of *S. flavescens*, because both of them belong to non-swollen bacteroids.
However, they differ from the swollen bacteroids of peanut that some of the genes
were up-regulated significantly, especially for the MurJ (MviN) and FtsZ
(WN72_33760). MurJ is an inner membrane protein responsible for the
murein/peptidoglycan biosynthesis by transferring the lipid II into the periplasm from

the cytoplasm in gram-negative cells (38) and is necessary for *Sinorhizobium meliloti*
(39). The MurJ is also required for cell wall biogenesis in gram-positive cells (40).
MviN is required for maintenance of cellular shape and integrity (41). The role of
MurJ (MviN) in development and differentiation of swollen bacteroids in peanut
nodules remains to be elucidated in the future. However, these differences imply that
the cell division genes in rhizobia might be related to the bacteroid differentiation, and
the host plant might affect the expression of these genes.

Cell division protein, FtsZ, is the target of nodule-specific cysteine-rich (NCR)
peptides and the interaction between them caused cell elongation (42, 43). Zhao *et al.*
(44) found that the importance of the FtsZ2 in bacteroid differentiation, and their
function could be inhibited by exogenous MreB protein. The higher expression of *ftsZ*
genes of four cell division proteins in peanut bacteroids might be another determinant
factor for the formation of swollen bacteroids. Lytic transglycosylase domain
(RL4716 = RL_RS24265) is required for proper cell envelope function in *Rhizobium*
*leguminosarum*, while the mutation of the RL4716 domain only influences the shape
of free-living cells, but did not impact the nodulation and nitrogen deficiency with
host plant pea (45). However, the shape and differentiation of RL4716 mutant
bacteroid in pea nodule were not checked.

The rhizobial BacA protein is an ABC transporter ATP-binding protein/permease,
firstly identified in *Sinorhizobium meliloti* by Glazebrook *et al.* (46) as being crucial
for bacteroid differentiation in alfalfa (*Medicago sativa*). BclA, homological to BacA,
was found in symbiotic *Bradyrhizobium* spp. associated with *Aeschynomene* plants
(47, 48). The BclA in *Bradyrhizobium* spp. strains ORS278 and ORS185 are found to
be essential for bacterial differentiation in symbiosis with *Aeschynomene* plants (48).
While *Bradyrhizobium diazoefficiens* USDA110 does not require *bclA* gene for
nitrogen fixing symbiosis with *A. afraspera* (48). In the *Mesorhizobium*
*huakuii*-*Astragalus sinicus* partner, protein BacA is indispensable for successful

establishment of the symbiotic system (49), similar to that in *S. meliloti* (46, 50). On
the contrary, *bacA* gene homolog, mlr7400, in *Mesorhizobium loti* MAFF303099 is
dispensable for symbiosis with *Lotus japonicus* (51, 52). In strain CCBAU 051107, it
contains a gene (WN72_20615) which has 60% protein homology to the BclA protein
(H0RSU6). The expression of WN72_20165 gene in free-living cell was about
185.146 (FPKM values), while it decreased to 54.2544 in bacteroids of peanut. The
function of the WN72_20165 protein in determination of bacteroid development
inside peanut nodule will be checked later as peanut nodule has many NCR peptides
(53) which may function on the differentiation of bacteroids.

Peng *et al.* (54) compared the transcriptome profiles of nodulating and non-nodulating
sister inbred peanut lines inoculated with *Bradyrhizobium* sp. and found some DEGs
involved in pathways of known symbiotic signaling interchange, and they also
discovered many other novel and/or functionally unknown genes related with plant
defense systems, hormone biosynthesis and response. In a most recent study, Raul *et*
*al.* focused on the transcriptome of peanut (53), but they did not care about the
metabolism of bacteroids inside the peanut nodules. Similarly, Han *et al.* (55) and Wei
*et al.* (56) determined alkaloids and flavonoids in *S. flavescens* by combining
metabolome and transcriptome, while they did not consider the nodulation. Our
current studies may close up the gap of these two kinds of studies, and in the future,
dual transcriptomes analyses combining the two symbiotic partners are needed to
explore the interaction and exchange of them in detail.

**Conclusion**

In our current studies, we compared the differences in morphology and gene
expression spectrums of the bacteroids in two host plants, peanut and *S. flavescens*,
nodulating by the same rhizobial strain *Bradyrhizobium arachidis* CCBAU 051107.
Bacteroids in peanut determinate nodules are spherical shape and swollen type, while
in the indeterminate nodules of *S. flavescens* they are nonswollen rod shaped.

Differences of gene expression of these two kinds of bacteroids are found mainly in
their nitrogen fixing, urea utilization pathway, peptidoglycan biosynthesis, cell
division, consistent with their shape, nitrogen fixing efficiency inside their host plants.
These results evidenced that the host plants had extensive effects on the
differentiation and metabolism of the microsymbionts. This is the first time to explore
the physiology and metabolism of bacteroids in peanut and *S. flavescens* and will be
helpful for us to promote or regulate the symbiotic nitrogen in these two kinds of
important legumes.

**Acknowledgments**

This work was funded mainly by National Natural Science Foundation of China (No.
31770039) to Wen Feng Chen. Thanks to Master Dan Zhang for collecting the strain
CCBAU 051107, later was identified as novel species *Bradyrhizobium arachidis* by
Ms. Rui Wang, from the peanut nodule samples grown in the field (the jobs were
organized by Dr. Xin Hua Sui). Appreciation to Ms. Shang Ying Wu, Mr. Yu Long
Wang and Prof. Bao Lin Guo for providing the *S. flavescens* (Kushen in Chinese)
seeds.

**Figure legends**

**Fig. 1. Root nodule, bacteroids, ARA and validation of RNA-seq vs. qRT-PCR.**

Root system, root nodule, nodule section and TEM of *A. hypogaea* (A1 to A4) and *S.*
*flavescens* (B1 to B4). NF, nitrogen fixation zone; MZ, meristem zone; PP,
electron-opaque polyphosphate inclusions; Bac, bacteroid; PBM, peribacteroid
membrane; PBS, peribacteroid space; PHB, poly- β -hydroxybutyrate. Acetylene
reduction assay (ARA) represented by production of C_2H_4 level of root nodules of *A.*
*hypogaea* (AH) and *S. flavescens* (SF)(C). Three stars (***) above the bars mean the
significant difference between the two samples of AH and SF using *t*-test. Validation
of RNA-seq results by qRT-PCR (D) using total 15 genes. F, free-living bacterium.

**Fig. 2. Volcano plot showing the up- and down-regulated expression genes in**
**bacteroids of *A. hypogaea* (A) and *S. flavescens* (B) vs. free-living bacteria.**

Selected differentially up-regulated expressed genes (*nifHDK, ureABEF*) are labeled
beside their dots. The plot was constructed by using function *ggscatter* of the “*ggpubr*”
package in R language.

**Fig. 3. The distribution of differentially expressed genes (DEGs) in COG**
**functional categories. A. Bacteroid of *S. flavescens* vs. free-living bacteria. B.**

Bacteroid of *A. hypogaea* vs. free-living bacteria. The plot was constructed by using
functions *ggplot, ggarrange* of the “*ggplot2*”, “*ggpubr*” packages in R language.

**Fig. 4. Top 30 up-regulated and down-regulated expression genes in bacteroids**
**of *A. hypogaea* (A) and *S. flavescens* (B) vs. free-living bacteria.** The up-regulated

expression genes were arranged in the upper half of the figure. The down-regulated
ones were shown in the latter half of the figure. The sun symbol (\odot) represents
components of nitrogenase, the diamond symbol (\diamond), the regulators of nitrogenase
activity, and the arrow (\rightarrow), co-factors. Common genes found in two kinds of

bacteroids were linked using lines. HP, hypothetical protein. The plot was constructed
by using function *diverging_lollipop_chart* of the “ggcharts” package in R language.

**Fig. 5. Differentially expressed genes (DEG) matched in KEGG database in**
**different pathway displayed by heatmap.** The data were from log₂ fold change of
DEG in bacteroids of *A. hypogaea* (AH) vs. free-living bacteria, and bacteroids of *S.*
*flavescens* (SF) vs. free-living bacteria. Legend was shown in subfigure K. The plot
was constructed by using function *pheatmap* of the “pheatmap” package in R
language. The plot was transferred into ppt by using function *topptx* of the “coffice”
package in R language to revise it manually.

**References**

[revised manuscript text omitted]

July 5, 2022

Dr. Wenfeng Chen
China Agricultural University
Yuanmingyuan Xilu No. 2
Beijing
China

Re: Spectrum01079-22 (Bacteroid development, transcriptome and symbiotic nitrogen-fixing comparison of Bradyrhizobium arachidis in nodules of Arachis hypogaea and Sophora flavescens)

Dear Dr. Wenfeng Chen:

Q: Thank you for submitting your manuscript to Microbiology Spectrum. While the reviewers agree that the manuscript describes a study that was well executed and scientifically sound, it is overly observational in nature. More needs to be said that speaks to the conclusions drawn and overall biological message taken from this work.

R: Thanks for your positive comments to our manuscript. The manuscript has been improved greatly according to the suggestions. Conclusions were extended according new analyses including gene expression of ammonium assimilation, Ni-Fe hydrogenase biosynthesis, bacteroid area etc. Discussion about the urea cycle were extended in the main text.

Q: When submitting the revised version of your paper, please provide (1) point-by-point responses to the issues raised by the reviewers as file type "Response to Reviewers," not in your cover letter, and (2) a PDF file that indicates the changes from the original submission (by highlighting or underlining the changes) as file type "Marked Up Manuscript - For Review Only". Please use this link to submit your revised manuscript - we strongly recommend that you submit your paper within the next 60 days or reach out to me. Detailed instructions on submitting your revised paper are below.

<https://spectrum.msubmit.net/cgi-bin/main.plex?el=A5QF3Bxug7A4FNyR7F6A9ftdUVsp3Bvyo4hV6MB01k6mtQZ>

R: Thanks for the guidance. Files about the point-by-point response, marked manuscript, and a clean manuscript were submitted. The instructions from the office were followed.

The ASM Journals program strives for constant improvement in our submission and publication process. Please tell us how we can improve your experience by taking this quick Author Survey.

R: Thanks for the instructions. Data has been submitted to GenBank and an accession number was written into the main text. It can be accessed now. Online supplement will be submitted to match the gene numbers used in this manuscript and the public data of the strain used. The manuscript is submitted on time.

The process of submission is smoothly and conveniently.

Sincerely,

Jannell Bazurto

Journals Department
Reviewer comments:

Reviewer #1 (Comments for the Author):

It is an outstanding study describing a unique field of knowledge making a behavioral comparison of bacteroids in two different plant types.

Here are some simple recommendations

Q:1- Please improve the English grammar in text, many of the suggestions have been imparted in the review file, but further work is needed.

R: Thanks for your kind comments. The English grammars in text were checked and revised. Almost all the tenses were changed to past tense.

Q: 2- Never start a sentence with an abbreviation or a number.

R: Thanks for your kind comments. The starts of sentences were checked and the abbr. or number were removed from the text. The changes could be found on lines 17, 255, 311, 364, 625.

Q: 3- Avoid unnecessary information as recommended in the commented file, it makes text boring for a reader.

R: Thanks for your comments. Unnecessary information especially in the methods were removed from the text.

Q: 4- It's recommended to use the past tense form while describing the methodology and results, where applicable.

R: Thanks for your kind comments. All the sentences have been checked and past tense was used.

Reviewer #2 (Public repository details (Required)):

Q: This study has done transcriptomics on nodule bacteroids of the two legumes peanut and *Sophora flavescens*. These are big data which need to be deposited at a public repository.

R: Yes, the big data has been deposited in NCBI with accession number PRJNA819203.

Reviewer #2 (Comments for the Author):

Q: The paper is strong technically. The authors have included very detailed methodologies and were honest in their approach.

R: Thanks for your kind comments.

Staff Comments:

Preparing Revision Guidelines

Q: To submit your modified manuscript, log onto the eJP submission site at <https://spectrum.msubmit.net/cgi-bin/main.plex>. Go to Author Tasks and click the appropriate manuscript title to begin the revision process. The information that you entered when you first submitted the paper will be displayed. Please update the information as necessary. Here are a few examples of required updates that authors must address:

- Point-by-point responses to the issues raised by the reviewers in a file named "Response to Reviewers," NOT IN YOUR COVER LETTER.

Done. A file was submitted.

- Upload a compare copy of the manuscript (without figures) as a "Marked-Up Manuscript" file.

Done. A marked-up-manuscript was submitted.

- Each figure must be uploaded as a separate file, and any multipanel figures must be assembled into one file.

Done. Each figure was a separate file.

- Manuscript: A .DOC version of the revised manuscript

Done. A .DOC version of the revised manuscript was submitted.

- Figures: Editable, high-resolution, individual figure files are required at revision, TIFF or EPS files are preferred

Done. EPS files were submitted.

Q: For complete guidelines on revision requirements, please see the journal Submission and Review Process requirements at <https://journals.asm.org/journal/Spectrum/submission-review-process>. Submissions of a paper that does not conform to Microbiology Spectrum guidelines will delay acceptance of your manuscript. "

R: I will submit the revised manuscript using the systems according to the guidelines.

Q: Please return the manuscript within 60 days; if you cannot complete the modification within this time period, please contact me. If you do not wish to modify the manuscript and prefer to submit it to another journal, please notify me of your decision immediately so that the manuscript may be formally withdrawn from consideration by Microbiology Spectrum.

R: I will submit it within 60 days.

If your manuscript is accepted for publication, you will be contacted separately about payment when the proofs are issued; please follow the instructions in that e-mail. Arrangements for payment must be made before your article is published. For a complete list of Publication Fees, including supplemental material costs, please visit our website.

R: Ok, I will do that as arranged.

Q: Corresponding authors may join or renew ASM membership to obtain discounts on publication fees. Need to upgrade your membership level? Please contact Customer Service at Service@asmusa.org.

R: Ok, I will join the membership.

Review 2

Q: The manuscript titled "Bacteroid development, transcriptome and symbiotic nitrogen-fixing comparison of *Bradyrhizobium arachidis* in nodules of *Arachis*

hypogaea and Sophora flavescens” by. Chen et al. describes the comparative transcriptomics of Bradyrhizobium arachidis in the swollen bacteroids of Peanut and the non-swollen bacteroids of Sophora flavescens. The honest efforts of the authors are clearly visible but the manuscript is not up to the standard for publication in the current journal.

R: Thanks for your comments. The manuscript has been improved greatly and submitted it to the journal again. Please consider it.

Below are my comments:

Q: One major comment I have is the lack of a clear question in the manuscript. Generating a big data and just describing it serves very little purpose towards the advancement of the understanding of particular subject. The authors have generated transcriptomes but have not dug deep enough to come up with a strong question that can be addressed effectively. The manuscript currently reads very flat and the observations are pretty general. The overall language of the manuscript needs to be improved considerably. There are several typos throughout as well as mis constructed sentences.

R: Thanks for your comments. In the revised manuscript, the importance and advancement were emphasized and extra figure, description, discussion were added to the text. The language was improved greatly. And some typos were found and removed in the revised manuscript.

Q: Fig1: the authors are comparing the nodule sections of the two legumes. The central theme of the paper is swollen bacteroids vs non-swollen bacteroids. But in this figure the authors do not show any images of the bacteroids of *S. flavescens*. The TEM images do not clearly reveal the shape of the bacteroids. Also, the image qualities are not up to the mark. The authors mention about nodule count but that needs to be represented graphically. Barplots are not a good way of representing the acetylene reduction data or any data as a matter of fact. The nodule comparison sections lack new information. The first three segments could be combined under a

single strong heading. Currently, these sections read as disconnected observations.

R: Thanks for your comments. A new figure was used to replace previous image of bacteroids of *S. flavescens* in Fig. 1B4. Now the bacteroids were clearly seen from the figure. Clearly, the bacteroids of *S. flavescens* were rod.

The nodule numbers were supplied and added to the text (line 266-267).

The Y-axis of the barplot for ARA data was revised to production of C₂H₄ so it would be appropriate to represent the acetylene reduction for the production of C₂H₄ was the characterization of nitrogenase activity.

More information about the nodule section were supplied including the size of the bacteroids (Fig. 1C) and the area of bacteroid section (text, line 294-295).

And the three segments have been combined into one section.

Q: The figure showing validation of the RNA seq data with qRT PCR on selected 15 genes is not proper. One cannot make out the genes used to validate the data. The authors should go for a much simpler yet effective bar plot showing the distribution of the data points. (Fig.1D)

R: Thanks for your kind comments. That figure was removed from the text and a new bar plot showing the distribution of the data was added in Fig. 1E, 1F, 1G.

Q: Fig2: the dynamics of the *ureABE* genes have not been described properly in the text.

R: Thanks for your kind comments. The dynamics of *ure* genes were described in the text. The changes of *ure* genes were exhibited in heatmap of Fig. 5L. Clearly, the expressions of *ure* genes in bacteroids of *S. flavescens* were higher than those in bacteroids of peanut. More descriptions of urea-related pathway were stated in the text.

Q: Instead of showing the up and down regulations like this, the authors could include a flower plot that would indicate the common up-down regulated genes while the specific genes for each bacteroid kind.

R: Thanks for your kind comments. Two flower plots were prepared and showed in

the text in Fig. 2.

Q: Fig3: better visual representation. This figure really serves very little purpose.

Here also the text reads flat and no specific message is delivered.

R: Thanks for your kind comments. The legend for this Fig. 3 was improved and more information were provided as following:

Fig. 3. The distribution and numbers of differentially expressed genes (DEGs) in COG functional categories. A. Bacteroid of *S. flavescens* vs. free-living bacteria. **B.** Bacteroid of *A. hypogaea* vs. free-living bacteria. Gene expressions were down-regulated or up-regulated respectively in the two kinds of bacteroids and were classified into different COG categories compared with the free-living bacteria. Numbers of DEGs were counted. The plot was constructed using functions *ggplot*, *ggarrange* of the “*ggplot2*”, “*ggpubr*” packages in R language.

October 3, 2022

Dr. Wenfeng Chen
China Agricultural University
Yuanmingyuan Xilu No. 2
Beijing
China

Re: Spectrum01079-22R1 (Bacteroid development, transcriptome and symbiotic nitrogen-fixing comparison of *Bradyrhizobium arachidis* in nodules of peanut (*Arachis hypogaea*) and medicinal legume *Sophora flavescens*)

Dear Dr. Wenfeng Chen:

While I feel the paper is close to acceptance at Spectrum, it would be in your best interests to improve the writing. I recommend that you ask a colleague of yours who is a native English speaker to read and provide you some feedback on the writing. You are also welcome to use one of the services here

<https://journals.asm.org/content/language-editing-services>

Link Not Available

Sincerely,

Jannell Bazurto

Journals Department
Reviewer comments:

Staff Comments:

Preparing Revision Guidelines

Please return the manuscript within 60 days; if you cannot complete the modification within this time period, please contact me. If you do not wish to modify the manuscript and prefer to submit it to another journal, please notify me of your decision immediately so that the manuscript may be formally withdrawn from consideration by Microbiology Spectrum.

December 29, 2022

Dr. Wenfeng Chen
China Agricultural University
Department of Microbiology and Immunology
Yuanmingyuan Xilu No. 2
Beijing
China

Re: Spectrum01079-22R2 (Bacteroid development, transcriptome and symbiotic nitrogen-fixing comparison of *Bradyrhizobium arachidis* in nodules of peanut (*Arachis hypogaea*) and medicinal legume *Sophora flavescens*)

Dear Dr. Wenfeng Chen:

Your manuscript has been accepted, and I am forwarding it to the ASM Journals Department for publication. You will be notified when your proofs are ready to be viewed.

Sincerely,

Jannell Bazurto
Editor, Microbiology Spectrum